# Understanding degraded speech leads to perceptual gating of a brainstem reflex in human listeners

**Heivet Hernández-Pérez**[1]*, **Jason Mikiel-Hunter**[1], **David McAlpine**[1], **Sumitrajit Dhar**[2], **Sriram Boothalingam**[3], **Jessica J. M. Monaghan**[1,4], **Catherine M. McMahon**[1]

**1** Department of Linguistics, The Australian Hearing Hub, Macquarie University, Sydney, Australia, **2** Department of Communication Sciences and Disorders, Northwestern University, Evanston, Illinois, United States of America, **3** University of Wisconsin-Madison, Madison, Wisconsin, United States of America, **4** National Acoustic Laboratories, Sydney, Australia

☉ These authors contributed equally to this work.

\* heivet.hernandez-perez@mq.edu.au

**Data Availability Statement:** The underlying data can be found in https://doi.org/10.5061/dryad.3ffbg79fw.

## Abstract

The ability to navigate "cocktail party" situations by focusing on sounds of interest over irrelevant, background sounds is often considered in terms of cortical mechanisms. However, subcortical circuits such as the pathway underlying the medial olivocochlear (MOC) reflex modulate the activity of the inner ear itself, supporting the extraction of salient features from auditory scene prior to any cortical processing. To understand the contribution of auditory subcortical nuclei and the cochlea in complex listening tasks, we made physiological recordings along the auditory pathway while listeners engaged in detecting non(sense) words in lists of words. Both naturally spoken and intrinsically noisy, vocoded speech—filtering that mimics processing by a cochlear implant (CI)—significantly activated the MOC reflex, but this was not the case for speech in background noise, which more engaged midbrain and cortical resources. A model of the initial stages of auditory processing reproduced specific effects of each form of speech degradation, providing a rationale for goal-directed gating of the MOC reflex based on enhancing the representation of the energy envelope of the acoustic waveform. Our data reveal the coexistence of 2 strategies in the auditory system that may facilitate speech understanding in situations where the signal is either intrinsically degraded or masked by extrinsic acoustic energy. Whereas intrinsically degraded streams recruit the MOC reflex to improve representation of speech cues peripherally, extrinsically masked streams rely more on higher auditory centres to denoise signals.

## Introduction

Robust cocktail party listening, the ability to focus on a single talker in a background of simultaneous, overlapping conversations, is critical to human communication and a long-sought goal of hearing technologies [1,2]. Problems listening in background noise are a key complaint of many listeners with even mild hearing loss and a stated factor in the non-use and non-uptake of hearing

**Funding:** H.H.P. was supported in this study by an International Macquarie University Excellence Scholarship (https://www.mq.edu.au/research/phd-and-research-degrees/scholarships/scholarship-search/data/international-hdr-main-scholarship-round) and the The HEARing Cooperative Research Centre (https://www.hearingcrc.org/) J.M.H. was supported in this study by an Australian Research Council Laureate Fellowship (FL 160100108) awarded to D.M (https://www.arc.gov.au/grants/discovery-program/australian-laureate-fellowships).The funders had no role in study design, data collection and analysis, decision to publish, or preparation of the manuscript.

**Competing interests:** The authors have declared that no competing interests exist.

**Abbreviations:** ABR, auditory brainstem response; AC, auditory cortex; AM, amplitude-modulated; AN, auditory nerve; BN, babble noise; CEOAE, click-evoked otoacoustic emission; CI, cochlear implant; CNC, consonant–nucleus–consonant; DPOAE, distortion product otoacoustic emission; DRNL, dual-resonance nonlinear; ERP, event-related potential; FPL, forward equivalent pressure level; HSR, high spontaneous rate; IC, inferior colliculus; LPC, late positivity complex; LSB, least significant bit; LSR, low spontaneous rate; MAP_BS, Matlab Auditory Periphery and Brainstem; MEMR, middle ear muscle reflex; MOC, medial olivocochlear; n.s., nonsignificant; OAE, otoacoustic emission; OHC, outer hair cell; rANOVA, repeated measures ANOVA; SNR, signal-to-noise ratio; SSN, speech-shaped noise; TFS, temporal fine structure.

devices [3–5]. However, despite their importance in everyday listening tasks and relevance to hearing impairment, physiological mechanisms that enhance attended speech remain poorly understood. In addition to local circuits in the auditory periphery and brainstem that have evolved to enhance automatically the neural representation of ecologically relevant sounds [6–8], it is likely that such a critical goal-directed behaviour as cocktail party listening also relies on top-down, cortically driven processes to emphasise perceptually relevant sounds and suppress those that are irrelevant [9,10]. Nevertheless, the specific role of bottom up and top-down mechanisms in complex listening tasks remain to be determined.

A potential mechanistic pathway supporting cocktail party listening is the auditory efferent system, whose multisynaptic connections extend from the auditory cortex (AC) to the inner ear [11–13]. In particular, reflexive activation by sound of fibres in the medial olivocochlear (MOC) reflex innervating the outer hair cells (OHCs—electromotile elements responsible for the cochlea's active amplifier) is known to reduce cochlear gain [14], increasing the overall dynamic range of the inner ear and facilitating sound encoding in high levels of background noise [15].

MOC fibres (ipsilateral and contralateral to each ear) originate in medial divisions of the superior olivary complex in the brainstem and synapse on the basal aspects of OHCs to modulate directly mechanical [16,17] and indirectly neural sensitivity to sound [18,19]. MOC neurons are also innervated by descending fibres from AC and midbrain neurons [11,20,21], providing a potential means by which the MOC reflex might be gated perceptually, either by directly exciting/inhibiting MOC fibres or by modulating their stimulus-evoked reflexive activity [22–25]. Although it has been speculated that changes in cochlear gain mediated by the MOC reflex might enhance speech coding in background noise [26–28], its role in reducing cochlear gain during goal-directed listening in normal-hearing human listeners (i.e., those with physiologically normal OHCs) remains unclear. In particular, it is unknown under which conditions the MOC reflex is active, including whether listeners must actively be engaged in a listening task for this to occur [27,29,30].

MOC reflex–mediated changes in cochlear gain can be assessed by measuring otoacoustic emissions (OAEs), energy generated by the active OHCs and measured noninvasively as sound from the ear canal [31]. When transient sounds such as clicks are delivered to one ear in the presence of noise in the opposite ear, OAE amplitudes are expected to be reduced, reflecting increased MOC reflex activity [32]. However, the extent to which OAEs are suppressed has been reported as either positively [29,33,34], negatively [27,35] or even uncorrelated [36,37] with the performance in speech-in-noise tasks. Modulation of cochlear gain through the MOC reflex could depend on factors such as task difficulty or relevance (e.g., speech versus non-speech tasks) and even methodological differences in the way in which inner ear signatures such as OAEs are recorded and analysed [29,38].

Here, we sought to determine whether cochlear gain is modulated in a task-dependent manner by selective recruitment of the MOC reflex. If the MOC reflex is sensitive to goal-directed control and improves understanding of degraded speech, then increases in task difficulty should be accompanied by reduced cochlear gain. We therefore assessed the extent to which the MOC reflex controls cochlear gain in active versus passive listening, i.e., when participants were required to attend to speech stimuli in order to complete a listening task compared to when they were not required to attend and instead watched a silent, non-subtitled film. In order to manipulate task difficulty, we employed speech sounds in background noise —stimuli traditionally used to evoke the MOC reflex [39–41]—and noise vocoding of "natural," clean speech—filtering that mimics processing by a cochlear implant (CI) [42]. Unlike speech in noise, noise-vocoded speech allows for manipulation of intelligibility without the addition of spectrally broadband acoustic energy that intrinsically activates the MOC reflex

[43–46]. Noise-vocoded speech should therefore enable better detection of any modulation of the MOC reflex by perceptual gating as task difficulty increases.

Physiological recordings in the central auditory pathway, including brainstem, midbrain, and cortical responses, were made while listeners performed an active listening task (detecting non-words in a string of Australian-English words and non-words). Importantly, our experimental paradigm was designed to maintain fixed levels of task difficulty. This allowed us to preserve comparable task relevance across different speech manipulations and avoid confounding effects of task difficulty on attention-gated activation of the MOC reflex. Additionally, visual and auditory scenes were identical across conditions to control for differences in alertness between active and passive listening.

We found that when task difficulty was maintained across speech manipulations, measures of hearing function at the level of the cochlea, brainstem, midbrain, and cortex were modulated depending on the type of degradation applied to speech sounds and on whether speech was actively attended. Specifically, the MOC reflex, assessed in terms of the suppression of click-evoked otoacoustic emissions (CEOAEs), was activated by noise-vocoded speech—an intrinsically degraded speech signal—but not by otherwise "natural" speech presented in either babble noise (BN; i.e., noise consisting of 8 talkers) or speech-shaped noise (SSN; i.e., noise sharing the long-term average spectrum of our speech corpus). Further, neural activity in the auditory midbrain was significantly increased in active versus passive listening for speech in BN and SSN, but not for noise-vocoded speech. This increase was associated with elevated cortical markers of listening effort for the speech-in-noise conditions. A model of the peripheral auditory system and its processes, including the MOC reflex, confirmed the stimulus-dependent role of the MOC reflex in enhancing neural coding of the speech envelope—a feature strongly correlated with the decoding and understanding of speech [47–51]. Our data suggest that otherwise identical performance in active listening tasks may invoke quite different efferent circuits, requiring not only different levels, but also different kinds, of listening effort, depending on the type of stimulus degradation experienced.

## Results

### Maintaining task relevance across speech manipulations requires isoperformance

We assessed speech intelligibility—specifically the ability to discriminate between Australian-English words and non-words—when speech was degraded by 3 different manipulations: noise vocoding the entire speech waveform; adding 8-talker BN to "natural" speech in quiet; or adding SSN to "natural" speech. Participants were asked to make this lexical decision (by means of a button press) when they heard a non-word in a string of words and non-words (Fig 1A and 1C).

Distinct levels of task difficulty were achieved by altering either the number of noise-vocoded channels—16 (Voc16), 12 (Voc12), and 8 (Voc8) channels—or by altering the signal-to-noise ratio (SNR) when speech was masked by BN—+10 (BN10) and +5 (BN5) dB SNR—or SSN—+8 (SSN8) and +3 (SSN3) dB SNR—(Fig 1D). This modulation of task difficulty was statistically confirmed in all 56 listeners ($n = 27$ in the vocoded condition and $n = 29$ in the 2 masked conditions) who showed consistently better performance—a higher rate of detecting non-words—in the less degraded conditions, i.e., more vocoded channels or higher SNRs in the masked manipulations: repeated measures ANOVA (rANOVA) [vocoded: [$F_{(3, 78)} = 70.92$, $p = 0.0001$, $\eta^2 = 0.73$]; BN: [$F_{(3, 78)} = 70.92$, $p = 0.0001$, $\eta^2 = 0.80$]; and SSN: [$F_{(2, 56)} = 86.23$, $p = 0.0001$, $\eta^2 = 0.75$]; see Fig 1D for post hoc analysis with Bonferroni corrections

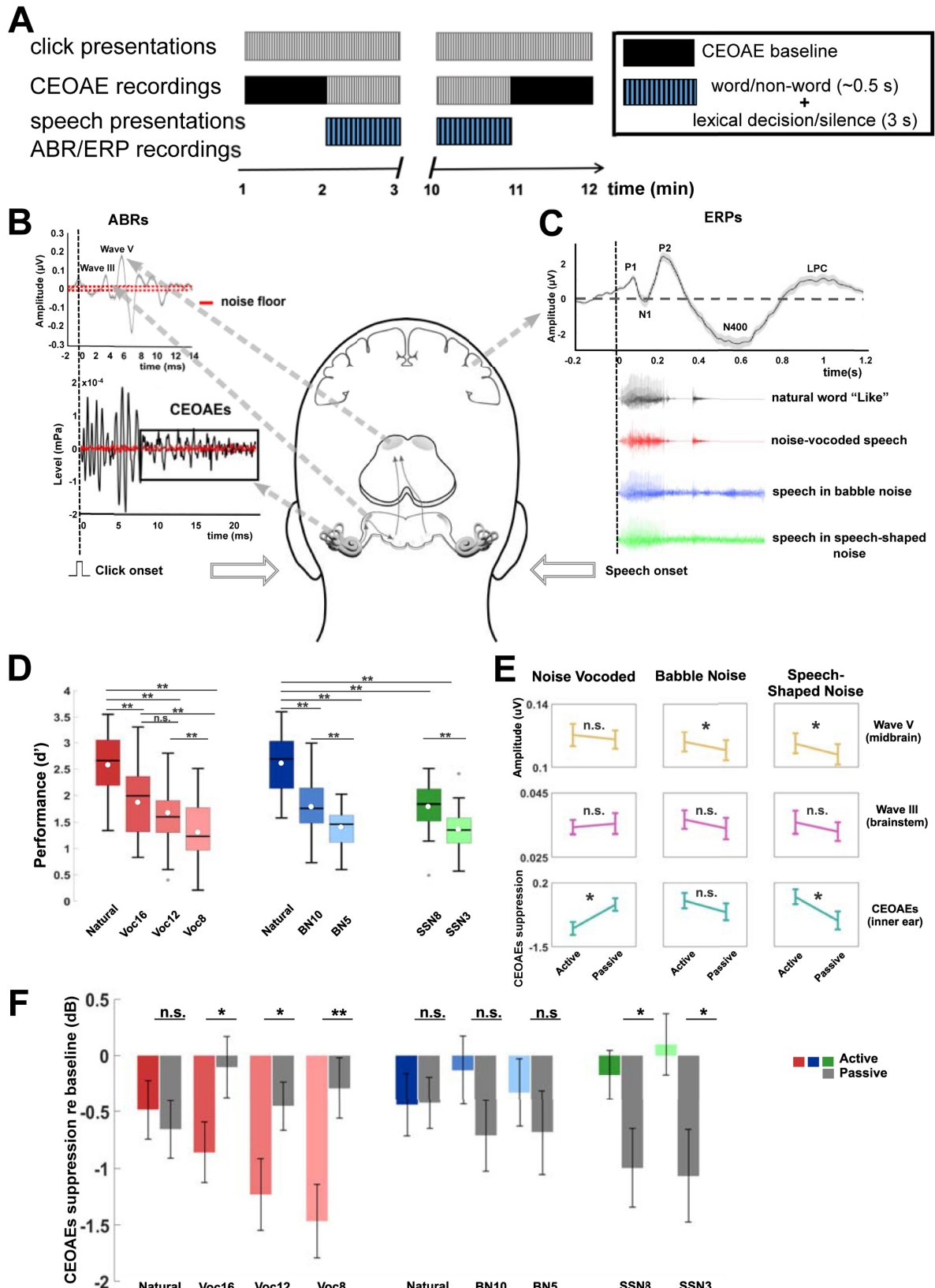

**Fig 1. Behavioural and physiological measurements during active and passive listening. (A)** Schematic representation of the experimental paradigm. Clicks were continuously presented for 12 minutes (grey-striped rectangles) in each experimental condition (e.g., active listening of natural speech). A 1-minute recording of baseline CEOAE magnitudes (black rectangles) was made at the beginning and at the end of each experimental condition. However, due to artefacts in CEOAE recordings, only the initial CEOAE baseline proved to be of sufficient quality for analysis. Speech tokens (blue and black-striped rectangles) were presented for 10 minutes to the ear contralateral to the ear receiving clicks. ABRs and ERPs were also recorded during this time frame. Following presentation of a word or non-word, participants had 3 seconds to make a lexical decision in the active listening condition (i.e., to determine whether an utterance was a word or a non-word) by pressing a button, or they remained silent in the passive condition by ignoring all auditory stimuli while watching a movie. **(B)** Schematic shows how click stimuli generated both CEOAEs (analysis window enclosed in rectangle) as well as brainstem activity (ABRs). **(C)** Corresponds to the time course of natural and degraded speech stimuli in relation to cortical activity (ERPs). **(D)** Performance during the lexical decision task. Mean d' (measure of accuracy calculated as Z (correct responses) − Z (false alarm) [i.e., Z (correct responses) = NORMSINV (correct responses)] denoted as white circles ($n$ = 27 in the noise-vocoded condition and $n$ = 29 in the 2 masked conditions). The horizontal line denotes the median. Upper and lower limits of the boxplot represent first (q1) and third (q3) quartiles, respectively, while whiskers denote the upper (q3+1.5*IQR) and lower bounds (q1-1.5*IQR) of the data where IQR is the interquartile range calculated as (q3-q1). Post hoc pairwise comparisons showed that highest performance was always achieved for natural speech compared to noise-vocoded, BN, and SSN manipulations (Bonferroni corrected $p$ = 0.001). Performance was moderately high (statistically lower than natural speech but higher than Voc8, BN5, and SSN3) for Voc16/Voc12 (combined due to n.s. differences, Bonferroni corrected $p$ = 0.11) as well as for BN10 and SSN8, respectively. The lowest level of performance was predictably observed for Voc8, BN5, and SSN3 ($p$ = 0.001). **(E)** CEOAEs and ABRs collapsed (rANOVA main effect of conditions (active and passive)). CEOAEs and ABRs are reported as means ± SEM. (**Bonferroni corrected $p < 0.01$; *Bonferroni corrected $p < 0.05$). **(F)** CEOAE suppression. The figure shows mean CEOAE magnitude (dB SPL) changes relative to the baseline for all conditions where negative values represent an increase in suppression (activation of MOC reflex). The underlying data can be found in https://doi.org/10.5061/dryad.3ffbg79fw. ABR, auditory brain stem response; BN, babble noise; CEOAE, click-evoked otoacoustic emission; ERP, event-related potential; IQR, interquartile range; n.s., nonsignificant; rANOVA, repeated measures ANOVA; SNN, speech-shaped noise.

(6 multiple comparisons for the noise-vocoded experiment and 3 for BN and SSN manipulations).

Isoperformance was achieved across speech manipulations, with best performance observed in the 2 natural speech conditions—one employed during the noise vocoding experiment and the other during the masking experiments: one-way ANOVA [F (1, 54) = 0.43, $p$ = 0.84, η2 = 0.001]. A moderate and similar level of performance (significantly lower than performance for natural speech) was achieved across Voc16/Voc12 (Voc16 versus Voc12: nonsignificant [n.s.] post hoc $t$ test: [t (26) = 2.53, $p$ = 0.11, d = 0.34]), BN10, and SSN8 conditions: one-way ANOVA: [F (3, 108) = 0.67, $p$ = 0.57, η2 = 0.018]. The poorest performance, significantly lower than the high and moderate performance levels, was observed for Voc8, BN5, and SSN3: one-way ANOVA: F (3, 81) = 0.07, $p$ = 0.72, η2 = 0.008].

Increasing task difficulty has been linked to the allocation of auditory attention and cognitive resources towards the task itself [52]. We, therefore, employed the discrete and matching levels of task difficulty across speech manipulations as a proxy for the required auditory attention.

## MOC reflex is modulated by task engagement in a stimulus-dependent manner

To determine whether auditory attention modulates cochlear gain via the auditory efferent system in a task-dependent manner, we assessed the effect of active versus passive listening and speech manipulation on the activity of the inner ear. CEOAEs were recorded continuously while participants either actively performed the lexical task or passively listened to the same corpus of speech tokens (Fig 1A). We first confirmed that CEOAE amplitudes were significantly reduced relative to a baseline measure (obtained in the absence of speech, Fig 1A) within each stimulus manipulation (planned $t$ test comparisons, S1 Table). CEOAEs were significantly reduced in magnitude when actively listening to natural speech and all noise-vocoded stimuli (natural: [t (24) = 2.33, $p$ = 0.03, d = 0.50]; Voc16: [t (23) = 3.40, $p$ = 0.002, d = 0.69]; Voc12: [t (24) = 3.98, $p$ = 0.001, d = 0.80] and Voc8: [t (25) = 5.14, $p$ = 0.001, d = 1.00]). Conversely, during passive listening, CEOAEs obtained during natural, but not noise-vocoded

speech were significantly smaller than baseline: [t (25) = 2.29, $p$ = 0.03, d = 0.44] (S1 Table). This was also true of CEOAEs recorded during the 2 masked conditions at all SNRs (natural: [t (26) = 2.17, $p$ = 0.04, d = 0.42]; BN10: [t (28) = 2.80, $p$ = 0.009, d = 0.52] and BN5: [t (28) = 2.36, $p$ = 0.02, d = 0.44]; SSN8: [t (28) = 3.37, $p$ = 0.002, d = 0.63] and SSN3: [t (28) = 3.50, $p$ = 0.002, d = 0.65]). This suggests that the MOC reflex is gated differently in active and passive listening, and by the different types of speech manipulation, despite listeners achieving isoper-formance across experimental conditions (i.e., comparable levels of lexical discrimination).

We calculated the reduction in CEOAEs between baseline and experimental conditions (CEOAE suppression, a proxy for activation of the MOC reflex) to quantify efferent control of cochlear gain in active and passive listening. For noise-vocoded speech, suppression of CEOAEs was significantly greater when participants were actively engaged in the lexical task compared to when they were asked to ignore the auditory stimuli: rANOVA: [F (1, 22) = 8.49, $p$ = 0.008, $\eta^2$ = 0.28] (Fig 1E and 1F). Moreover, we observed a significant interaction between conditions and stimulus type: [F (3, 66) = 2.80, $p$ = 0.046, $\eta^2$ = 0.12], indicating that the suppression of CEOAEs was stronger for all vocoded conditions in which listeners were required to make lexical decisions, compared to when they were not (Fig 1F)—Voc16: [t (23) = −2.16, $p$ = 0.04, d = 0.44]; Voc12: [t (24) = −2.19, $p$ = 0.038, d = 0.44] and Voc8: [t (25) = 3.51, $p$ = 0.002, d = 0.69]. Engagement in the task did not modulate CEOAE suppression for the natural speech condition: [t (24) = 0.62, $p$ = 0.54, d = 0.12].

By contrast, speech embedded in SSN elicited the opposite pattern of results to noise-vocoded speech (Fig 1E and 1F). The suppression of CEOAEs was significantly stronger during passive, compared to active, listening (Fig 1F): rANOVA: [F (1, 24) = 4.44, $p$ = 0.046, $\eta^2$ = 0.16], and we observed a significant interaction between condition and stimulus type: [F (2, 48) = 4.67, $p$ = 0.014, $\eta^2$ = 0.16] for both SNRs: SSN8 [t (27) = 2.71, $p$ = 0.01, d = 0.51] and SSN3 [t (28) = 2.67, $p$ = 0.012, d = 0.50]. We also observed a mild suppression of CEOAEs for speech masked by BN, with CEOAEs significantly smaller than their own baseline measures only during passive listening (shown in the planned $t$ test, S1 Table), but not when active and passive conditions were compared (Fig 1E and 1F) (rANOVA n.s.: F (1, 25) = 1.21, $p$ = 0.28, $\eta^2$ = 0.05). Cochlear gain was, therefore, suppressed during active listening of noise-vocoded speech, slightly but significantly suppressed during passive listening in BN, and strongly suppressed during passive listening in SSN. Together, our data suggest that the MOC reflex is modulated by task engagement and strongly depends on the way in which signals are degraded, including the type of noise used to mask speech.

## Auditory brainstem activity reflects changes in cochlear gain when listening to speech in noise

The effects of active versus passive listening on cochlear gain were evident in the activity of subcortical auditory centres when we simultaneously measured auditory brainstem responses (ABRs) to the same clicks used to evoke CEOAEs. Click-evoked ABRs largely reflect summed activity of higher-frequency regions of the cochlea (3 to 8 kHz [53,54]). However, as CEOAE suppression in the 1 to 2 kHz band is used here as a marker for MOC reflex activity across the entire length of the cochlea (see Materials and methods), we can relate observed changes in cochlear gain to amplitudes of ABR waves.

Click-evoked ABRs—measured during presentation of speech in noise—showed similar effects to those observed for CEOAE measurements. Specifically, in both masked conditions, wave V of the ABR—corresponding to neural activity generated in the midbrain nucleus of the inferior colliculus (IC)—was significantly enhanced in the active, compared to the passive, listening condition (Fig 1E) (speech in BN: [F (1, 26) = 5.66, $p$ = 0.025, $\eta^2$ = 0.20] and SSN:

[F (1, 26) = 9.22, $p$ = 0.005, $\eta^2$ = 0.26]). No changes in brainstem or midbrain activity were observed between active and passive listening of noise-vocoded speech.

To exclude this stimulus-dependent pattern of inner ear and brainstem responses arising from intrinsic differences in the 2 populations of listeners tested (noise-vocoded versus masked speech experiments), we compared CEOAE suppression as well as the amplitude of ABR waves between the 2 groups for active and passive listening of natural speech. No statistical differences were observed for either active or passive listening between the 2 groups (active natural condition: suppression of CEOAEs [t (23) = −0.21, $p$ = 0.83, d = −0.04; wave III [t (23) = −0.45, $p$ = 0.65, d = 0.09]; wave V [t (23) = 0.09, $p$ = 0.93, d = 0.02]; passive natural condition: suppression of CEOAEs [t (24) = −0.36, $p$ = 0.72, d = 0.07]; wave III [t (24) = −0.16, $p$ = 0.88, d = 0.03]; wave V [t (26) = 0.40, $p$ = 0.69, d = 0.05]). We conclude from this that the differences observed in cochlear gain and auditory brainstem/midbrain activity can be attributed to the specific form of speech degradation.

Together with the effect on CEOAEs, these data suggest that the magnitude of auditory midbrain activity for the different speech manipulations reflects cochlear output. While this is evident for both listening conditions in masked speech, the similarity of ABR magnitudes in the midbrain for active and passive listening of noise-vocoded stimuli is indicative of feed-forward amplification that compensates for reduced cochlear gain during active listening. This highlights increased emphasis of peripheral processing for noise-vocoded, compared to noise-masked, speech and suggests that processing by higher-order auditory centres may be involved in decoding masked speech.

## Simulated MOC reflex improves the neural representation of noise-vocoded speech, but not speech in noise

Previous modelling studies have supported the ability of the MOC reflex to "unmask" signals in noise in the auditory nerve (AN) [55–62] and, therefore, would not provide a suitable rationale for the absence of CEOAE suppression (i.e., a lack of MOC reflex activity) when participants actively listened to speech in BN or SSN (Fig 1E and 1F). To determine how the neural representation of degraded speech differs in the AN with and without the MOC reflex, and whether this might explain the stimulus dependence of CEOAE suppression in Fig 1E and 1F, we implemented a model of the initial auditory stages (outer, middle, and inner ear with the AN) that includes an MOC reflex [58,63–65]. We focused on how the MOC reflex affects the encoding of the energy envelope of acoustic waveforms: considered critical to speech understanding (especially noise-vocoded speech [42,50]) [42,50,66,67], strongly correlated with the cortical tracking and decoding of speech [47–49], and the basis of several successful speech intelligibility models [51,68,69].

Here, we tested the hypothesis that efferent suppression of cochlear gain differentially impacts neural encoding of masked and noise-vocoded stimulus envelopes in AN fibres. Natural and degraded speech tokens (those generating lowest, isoperformance in the active task: Voc8; BN5; and SSN3; Fig 1C) were presented to the model at 75 dBA, with and without a fixed 15 dB attenuation generated by the MOC reflex (Fig 2A). Responses of 400 AN fibres with low spontaneous rate (LSR) and high thresholds were simulated in each of 30 frequency channels, logarithmically spaced between 0.1 and 4.5 kHz, forming the model's output (Fig 2A and 2B). We chose this type of AN fibre for our model because of their apparent critical role in processing sounds in high levels of background noise [70–72].

To assess how the energy envelope of degraded speech tokens was represented in the output of the population of AN fibres, both normal and polarity-inverted copies of each token were presented to the model for processing (Fig 2A and 2B). The polarity tolerant component,

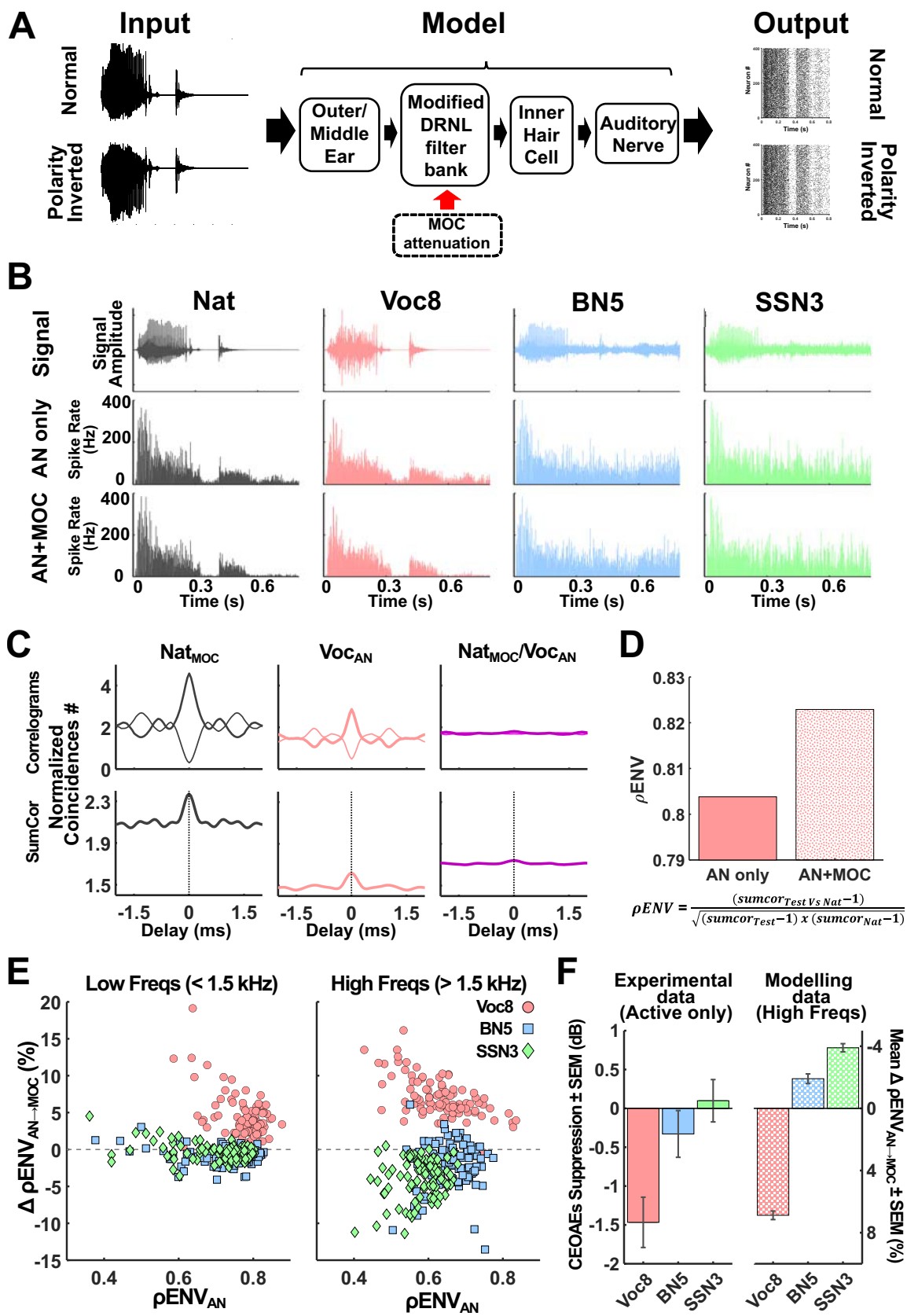

**Fig 2. Output of model of the initial auditory stages with and without inclusion of simulated MOC reflex.** **(A)** Schematic of model showing input stimulus and neural output. Normal and inverted polarity waveforms of words (here, the word "Like") were presented to the MAP_BS model that incorporates a cascade of stages from outer and middle ear to AN output. Attenuation of cochlear gain by the simulated MOC reflex was implemented at the "modified DRNL filter bank" stage [58,63]. Responses of 400 LSR fibres (shown here in the form of raster plots for normal and polarity-inverted waveforms of the word "Like") constituted the model output at the AN stage. **(B)** Presentation of natural and degraded versions of the word "Like" with and without simulated MOC reflex. Normal "Like" waveforms for natural (dark grey, far left), Voc8 (pink, second left), BN5 (light blue, second right), and SSN3 (green, far right) conditions are shown in the top row. Post-stimulus time histograms (average response of 400 fibres binned at 1 ms) were calculated for LSR AN fibres (characteristic frequency: 2.33 kHz) with (bottom row) and without (middle row) simulated MOC reflex. Including simulated MOC reflex reduced activity during quiet for natural condition (and Voc8, but less so) while maintaining high spiking rates at peak sound levels (e.g., at 0.075, 0.3, and 0.45ms). No changes in neural representation of signal were visually evident for BN5 and SSN3 "Like". **(C and D)** Quantifying ρENV for Voc8 "Like" without simulated MOC reflex in 2.33-kHz channel. *Sumcor* plots (bottom row, C) were generated by adding shuffled autocorrelograms (thick lines, top left/middle panels, C) or shuffled cross-correlograms (thick line, top right panel, C) to shuffled cross-polarity correlograms (thin lines, top row, C) to compare neural envelopes of the control condition, Naturally spoken "Like" with simulated MOC reflex (left/right columns, C), and the test condition, Voc8 "Like" without simulated MOC reflex (middle/right columns, C). ρENV for Voc8 "Like" without simulated MOC reflex (AN, *solid-pink bar*, D) was calculated from *sumcor* peaks in C [73]. Value of ρENV with simulated MOC reflex (AN+MOC, speckled pink bar, D) is also displayed. **(E and F)** Comparing ΔρENVs for 100 words after introduction of simulated MOC reflex. Mean percentage changes in ρENVs (calculated in 2 frequency bands: below and above 1.5 kHz) after adding simulated MOC reflex were plotted as a function of ρENV without simulated MOC reflex for degraded versions of 100 words (each symbol represents one word). ΔρENVs were positive for all Voc8 words except 1 (*pink circles*, E) (Max-Min ΔρENV for Voc8 for <1.5 kHz: +17.62 to −0.78%; Max-Min ΔρENV for Voc8 for >1.5 kHz: +16.14 to −0.62%), appearing largest for words with lowest ρENVs without simulated MOC reflex. This relationship was absent for BN5 (light blue squares, E) and SSN3 (green diamonds, E) words whose ΔρENV ranges spanned the baseline (Max-Min ΔρENV for BN5 for <1.5 kHz: +3.06 to −4.09%; Max-Min ΔρENV for BN5 for >1.5 kHz: +6.08 to −13.56%; Max-Min ΔρENV for SSN3 for <1.5 kHz: +4.52 to −3.58% Max-Min ΔρENV for SSN3 for >1.5 kHz: +0.50 to −11.39%). Progression of mean ΔρENVs (± SEM) for model data >1.5 kHz (checkerboard bars, right, F) mirrored that of the active listening task, CEOAE data (mean ± SEM) (solid colour bars, left, F). The underlying data can be found in https://doi.org/10.5061/dryad.3ffbg79fw. AN, auditory nerve; CEOAE, click-evoked otoacoustic emission; DRNL, dual resonance nonlinear; LSR, low spontaneous rate; MAP_BS, Matlab Auditory Periphery and Brainstem; MOC, medial olivocochlear.

associated with the stimulus envelope [50,73–77], was extracted from the responses of AN fibres using *sumcor* analysis (Fig 2C). We compared this similarity of neural envelopes between conditions by computing the neural cross-correlation coefficient, ρENV, in each frequency channel (Fig 2D) [50,73]. Values of ρENV, ranging from 0 to 1 for independent to identical neural envelopes, respectively, were calculated for the 3 speech manipulations with ($\rho ENV_{AN}$) and without ($\rho ENV_{MOC}$) the MOC reflex included in the model. In each case, the neural envelope for natural speech acted as the "control" template for comparison. "Control" simulations of natural speech were performed with the MOC reflex included, based on our observations that a steady CEOAE suppression—indicative of an active MOC reflex—occurred for natural speech experimentally (Fig 1F) and that neural envelopes for natural sentence stimuli were enhanced in model AN fibres with an MOC reflex present (S1 Fig).

Given the range of acoustic waveforms in our speech corpus, we included 100 words (50 stop/nonstop consonants) in our analysis (Fig 2E). Despite the diversity of speech tokens, the effects of including MOC reflex (on ρENV) were consistently dependent on the form of stimulus manipulation. Neural encoding of speech envelopes improved significantly with simulated MOC reflex for noise-vocoded words (pink circles, Fig 2E) (mean ΔρENV for Voc8 for freqs <1.5 kHz = +4.22 ± 0.30%, [$Z(99) = 8.66$, $p < 0.0001$, r = 0.86]; mean ΔρENV for Voc8 for freqs >1.5 kHz = + 6.88 ± 0.28%, [$Z(99) = 8.67$, $p < 0.0001$, r = 0.87]), with the largest enhancements observed for words with the lowest ρENV values in the absence of MOC reflex. By contrast, no such relationship was observed for words presented in BN (BN5, light blue squares, Fig 2E) or SSN (SSN3, green diamonds, Fig 2E). Moreover, envelope coding in both speech-in-noise conditions was significantly impaired, on average, when the MOC reflex was included (mean ΔρENV for BN5 for freqs <1.5 kHz = −0.89 ± 0.11, [$Z(99) = −6.67$, $p < 0.001$, r = 0.67]; mean ΔρENV for BN5 for freqs >1.5 kHz = −1.91 ± 0.31%, [$Z(99) = −5.704$, $p < 0.001$, r = 0.57]; mean ΔρENV for SSN3 for freqs <1.5 kHz = −2.62 ± 0.12, [$Z(99) =$

$-4.10$, $p = 0.001$, r $= 0.41$]; mean $\Delta\rho$ENV for SSN3 $>1.5$ kHz $= -3.90 \pm 0.26$, [Z (99) $= -8.66$, $p < 0.0001$, r $= 0.87$]). Given the apparent similarities between single polarity responses for corresponding noise-vocoded and natural stimuli in Fig 2B (i.e., Voc8$_{ANonly}$ versus Nat$_{ANonly}$ and Voc8$_{AN+MOC}$ versus Nat$_{AN+MOC}$, Fig 2B), we tested how values of $\Delta\rho$ENV were affected when the neural envelopes for degraded speech tokens were compared with their corresponding natural conditions (i.e., Voc8/BN5/SSN3$_{ANonly}$ versus Nat$_{ANonly}$ and Voc8/BN5/SSN3$_{AN+MOC}$ versus Nat$_{AN+MOC}$) (S2 Fig). The stimulus-specific effects we observed in Fig 2 not only remained with this new "control" template for $\rho$ENV$_{AN}$ (S2A Fig) but were also enhanced for all 3 speech manipulations (S2B Fig). The inclusion of the MOC reflex had similar stimulus-dependent effects in low ($<1.5$ kHz) and high ($>1.5$ kHz) frequency bands. However, given the increased importance carried by stimulus envelope for acoustic stimuli at high frequencies [66,78,79], only neural envelope encoding in the high-frequency band was considered in subsequent simulations and analysis.

The stimulus-specific changes in envelope encoding we observed were also evident, but with reduced magnitudes, when we lowered the fixed attenuation of the MOC from 15 dB to 10 dB (S3A Fig), demonstrating that manipulating the strength of the MOC reflex may provide a means of selectively enhancing or impairing neural encoding of the stimulus envelope. Increasing the SNR of the masked speech [i.e., from +5 SNR (BN5) to +10 SNR (BN10) for BN and from +3 SNR (SN3) to + 8 SNR (SN8) for SSN] not only diminished the detrimental effects of the MOC reflex on envelope encoding at the lower SNRs (S3B Fig), but also led to an overall improvement in envelope encoding with the MOC reflex for BN10 stimuli. This suggests that efferent feedback through the MOC reflex may be unable to enhance the neural representation of speech in background noise at low SNRs. By contrast, introducing the MOC reflex to the neural coding of stimuli with more noise-vocoded channels generated enhanced benefit (S3B Fig).

Although we assessed high-threshold fibres due to their importance at high sounds levels [70–72], the majority of AN fibres possess high spontaneous rate (HSR) with low-threshold (i.e., fibres that respond preferentially to low-intensity sounds but saturate at higher intensities [80–82]). These low-threshold fibres may not only play an important role in envelope processing of speech at low intensities but also contribute at high intensities thanks to their dynamic range adaptation and response fluctuations [78,83–87]. We therefore also assessed how these low-threshold, HSR fibres processed the most difficult stimulus degradations (Voc8, BN5, and SSN3; Fig 1D) in the presence and absence of the MOC reflex (S4 Fig). Similar to AN fibres with high thresholds, including the MOC reflex improved envelope encoding by low-threshold AN fibres for noise-vocoded speech and impaired it for speech masked by BN or SSN (S4A and S4B Fig). This is despite poorer dynamic range of low-threshold fibres at 75 dBA (normalised sound presentation level across manipulations) likely impacting their overall ability to encode the stimulus envelope.

We also examined the effects of efferent feedback on the encoding of temporal fine structure (TFS, the instantaneous pressure waveform of a sound)—a stimulus cue at low frequencies associated with speech understanding [50,88–92]—and observed a small, mean improvement in the model with the MOC reflex at low frequencies ($<1.5$ kHz) for the masked conditions with the lowest SNRs (i.e., BN5 and SSN3; S5 Fig). Although this improvement in TFS encoding is consistent with other studies whose simulations support a role for efferent unmasking in speech-in-noise processing [57–59,61], it cannot explain the lack of CEOAE suppression we observed experimentally for these, most difficult, masked speech tasks (Fig 1F).

Overall, the pattern of neural envelope enhancement observed in our model AN fibres (both low and high threshold) when the MOC reflex was introduced to the different stimulus degradations (right, Fig 2F) mirrored the observed suppression of CEOAEs for corresponding

active listening conditions (left, Fig 2F). Where activation of the MOC reflex was evident experimentally for noise-vocoded speech, enhancement of neural envelopes was observed when the same degraded stimuli were presented to the model with the MOC reflex present. This was not the case for active listening to masked speech. Here, we found no evidence that MOC activity was modulated by active listening to masked speech, and this was consistent with the poorer neural representations of the stimulus envelope when the MOC reflex was included in the model for these stimulus conditions.

## Cortical evoked potentials are enhanced when actively listening to speech in noise

The seeming lack of any contribution from the MOC reflex during active listening to speech masked by speech-like sounds (i.e., BN and SSN) compared to noise-vocoded speech suggests that other compensatory brain mechanisms must contribute to listening tasks if isoperformance is maintained across conditions. We therefore explored whether higher brain centres—providing top-down, perhaps attention-driven, enhancement of speech processing in background noise—contribute to maintaining isoperformance across the different speech degradations. In particular, the significant increase observed in wave V of the ABR for active speech-in-noise conditions suggests greater activity in the IC—the principal midbrain nucleus receiving efferent feedback from auditory cortical areas. Levels of cortical engagement might therefore be expected to differ depending on the form of speech manipulation, despite similar task performance.

To determine the degree of cortical engagement in the active listening task, we recorded cortical evoked potentials from all 56 participants—simultaneously with CEOAE and ABR measurements—using a 64-channel, EEG-recording system. Grand averages of event-related potentials (ERPs) to speech onset (Fig 3A, S6 Fig) for the most demanding speech manipulations [Voc8 (S6A Fig), BN5 (S6B Fig), and SSN3 (S6C Fig)] were analysed to test the hypothesis that greater cortical engagement occurred when listening to speech in background noise compared to noise-vocoded speech, despite their being matched in task difficulty.

We analysed early auditory cortical responses (P1 and N1 components, Fig 3A) that are largely influenced by acoustic features of the stimulus such as intensity and spectral content [98,99]. Noise-vocoded words elicited well-defined P1 and N1 components compared to the speech-in-noise conditions (Fig 3A), despite words and noises having similar onsets to noise-vocoded tokens. This likely reflects the relatively high precision of the envelope components of noise-vocoded speech at stimulus onset compared to the masked conditions in which the competing noises interfere with speech envelope, producing less precise neural responses [100–102].

Later ERP components, such as P2, N400 and the late positivity complex (LPC), are associated with speech- and task-specific, top-down (context-dependent) processes [103,104]. Speech masked by BN or SSN elicited significantly larger P2 components during active listening compared to the noise-vocoded condition, but not significantly different between themselves [F (2,79) = 5.08, $p$ = 0.008, $\eta^2$ = 0.11], post hoc with Bonferroni corrections for 3 multiple comparisons: [BN5 versus Voc8 ($p$ = 0.012, d = 0.78); SSN3 versus Voc8 ($p$ = 0.041, d = 0.69); BN5 versus SSN3 ($p$ = 1.00, d = 0.13)]. Similarly, the magnitude of the LPC—thought to reflect the involvement of cognitive resources including memorisation, understanding [105], and post-decision closure [106] during speech processing—differed significantly across conditions: [F (2,79) = 4.24, $p$ = 0.018, $\eta^2$ = 0.10]. Specifically, LPCs were greater during active listening to speech in BN compared to noise-vocoded speech ($p$ = 0.02, d = 0.85), with LPCs generated during active listening to speech in SSN intermediate to both, but not significantly

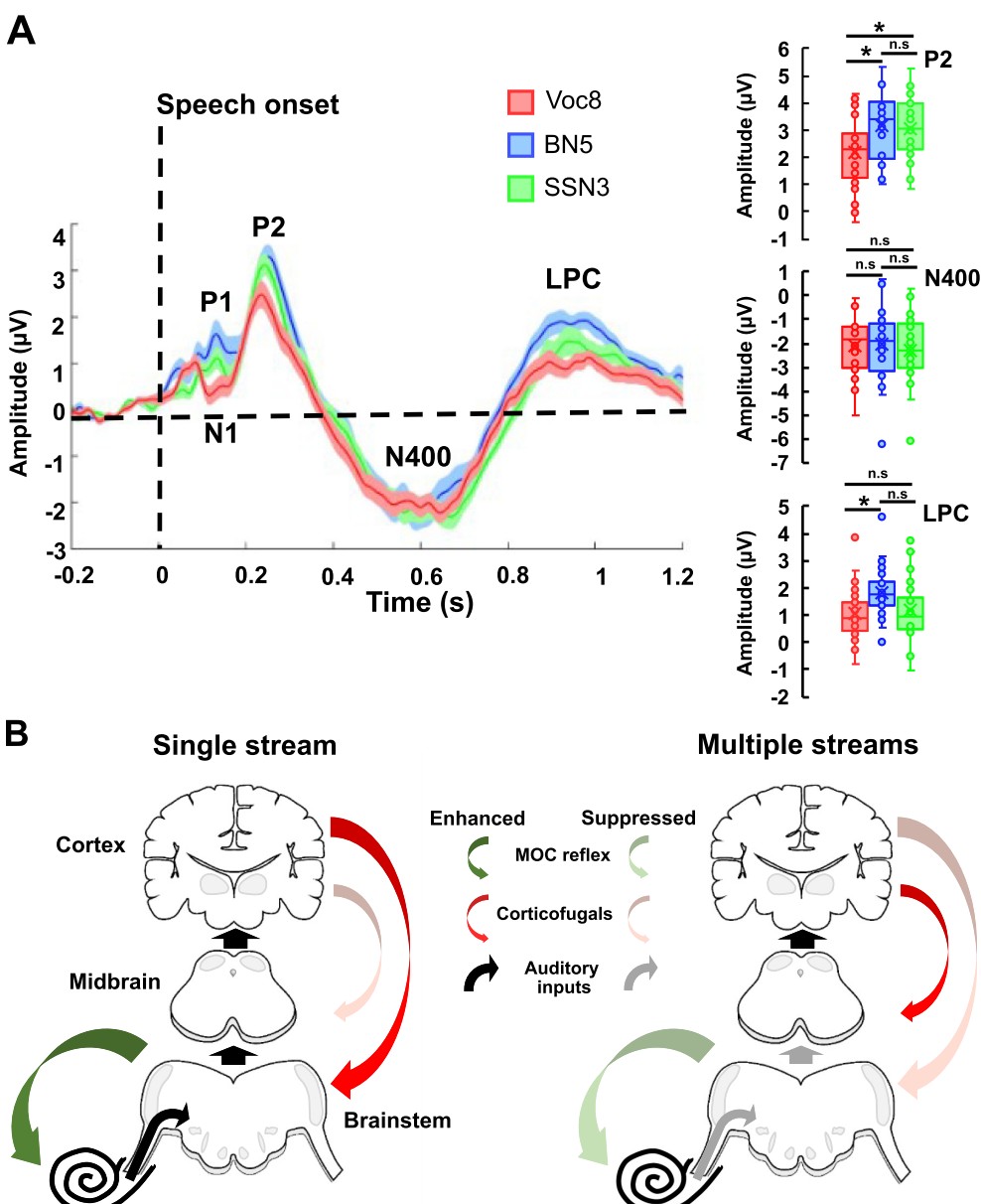

**Fig 3. Cortical activity and proposed mechanisms for active listening to noise-vocoded and masked speech. (A)** ERP components (from electrodes: FZ, F3, F4, CZ, C3, C4, TP7, TP8, T7, T8, PZ, P3, and P4) during the active listening of Voc8, BN5, and SSN3. Electrodes' selection was based on their relevance in attentional and language brain networks [93–97]. Thick lines and shaded areas represent mean and the SEM, respectively. Boxplots on the right show statistical comparisons between speech conditions for P2, N400, and LPC components. **(B)** Proposed auditory efferent mechanisms for speech processing. The "single stream" mechanism shows how degraded tokens such as noise-vocoded speech are processed in a mostly feed-forward manner (thick black arrows) (as should be the case for natural speech). The activation of the MOC reflex (dark green arrow) improves the AN representation of speech envelope (black arrow, from the cochlea shown as a spiral). This information passes up the auditory centres without much need to "denoise" the signal (represented as black arrows from the brainstem to midbrain to cortex). Given our observation that cochlear gain suppression increased with task difficulty, we included the possibility for enhanced MOC reflex drive from higher auditory regions via corticofugal connections (dark red arrows). By contrast, "multiple streams" such as speech in BN or SSN do not recruit the MOC reflex (light green arrow) because it negatively affects envelope encoding of speech signals (light grey arrows from cochlea–brainstem–midbrain). We therefore propose that corticofugal drive is suppressed to the MOC reflex (shaded red arrow), resulting in weaker MOC reflex activation (light green arrow). This leaves greater responsibility for speech signal extraction to the midbrain, cortex, and the efferent loop therein (corticofugal connections from AC to midbrain: dark red arrow). Both mechanisms ultimately lead to equal behavioural performance across speech conditions. The underlying data can be found in https://doi.org/10.5061/dryad.3ffbg79fw.

AC, auditory cortex; AN, auditory nerve; ERP, event-related potential; LPC, late positivity complex; MOC, medial olivocochlear.

different from either (Fig 3A). Consistent with effortful listening varying across speech manipulations even when isoperformance was maintained [107,108], the speech manipulation generating the clearest signature of cortical engagement was speech in a background of BN. This was considered the most difficult of the masked conditions (Fig 1D) where isoperformance was achieved at +5 dB SNR (BN5) compared to only +3 dB SNR for speech in SSN (SSN3). In contrast to P2 and the LPC, the N400 component of the ERP—associated with the processing of meaning [93]—did not differ between conditions [F (2,81) = 0.22, $p$ = 0.81, $\eta^2$ = 0.005]. This is unsurprising, given that participants were equally able to differentiate non-words in Voc8, BN5, and SSN3 conditions, given that isoperformance had been achieved.

Our ERP data are consistent with differential cortical contributions to the processing of noise-vocoded and masked speech, being larger in magnitude for speech manipulations in which the MOC reflex was less efficiently recruited, i.e., BN and SSN, and largest overall for the manipulation requiring most listening effort—speech in a background of multitalker babble. From the auditory periphery to the cortex, our data suggest that 2 different strategies coexist to achieve similar levels of performance when listening to single undegraded/degraded streams of speech compared to speech masked by additional noise (Fig 3B). The first involves enhanced sensitivity to energy fluctuations through recruitment of the MOC reflex to generate a central representation of the stimulus sufficient and necessary for speech intelligibility of single streams (Fig 3B, left panel). The second, implemented when processing speech in background noise (Fig 3B, right panel), preserves cochlear gain by gating off the MOC reflex off; suppression of cochlear gain by an active MOC reflex does not provide any added benefit to the encoding of the stimulus envelope in the periphery. This places the onus of "denoising" on midbrain and cortex auditory structures and processes including loops between them to maximise speech understanding for masked speech.

## Discussion

We assessed the role of attention in modulating the contribution of the cochlear efferent system in a lexical task—detecting non-words in a stream of words and non-words. Employing 3 speech manipulations to modulate task difficulty—noise vocoding of words, words masked by multitalker BN, or SSN (i.e., noise with the same long-term spectrum as speech)—we find that these manipulations differentially activate the MOC reflex to modulate cochlear gain. Activation of the cochlear efferent system also depends on whether listeners are performing the lexical task (active condition) or are not required to engage in the task and instead watch a silent, stop-motion film (passive condition). Specifically, with increasing task difficulty (i.e., fewer noise-vocoded channels), noise vocoding increasingly activates the MOC reflex in active, compared to passive, listening. The opposite is true for the 2 masked conditions, where words presented at increasingly lower SNRs more strongly activate the MOC reflex during passive, compared to active, listening. By adjusting parameters of the 3 speech manipulations—the number of noise-vocoded channels or the SNR for the speech-in-noise conditions, we find that lower MOC reflexive activity is accompanied by heightened cortical activation, possibly to maintain isoperformance in the task. A computational model incorporating efferent feedback to the inner ear demonstrates that improvements in neural representation of the amplitude envelope of sounds provides a rationale for either suppressing or maintaining the cochlear gain during the perception of noise-vocoded speech or speech in noise, respectively. Our data suggest that a network of brainstem and higher brain circuits maintains performance in active

listening tasks and that different components of this network, including reflexive circuits in the lower brainstem and the relative allocation of attentional resources, are differentially invoked depending on specific features of the listening environment.

## Attentional demands reveal differential recruitment of the MOC reflex

Our data highlight a categorical distinction between active and passive processing of single, degraded auditory streams (e.g., noise-vocoded speech) and parsing a complex acoustic scene to hear out a stream from multiple competing, spectrally similar sounds (multitalker babble and SSN). Specifically, task difficulty during active listening appears to modulate cochlear gain in a stimulus-specific manner. The reduction in cochlear gain with increasing task difficulty for noise-vocoded speech and, conversely, the preservation of cochlear gain when listening to speech in background noise, suggests that attentional resources might gate the MOC reflex differently depending on how speech is degraded. In contrast to active listening, when participants were asked to ignore the auditory stimuli and direct their attention to a silent film, the MOC reflex was gated in a direction consistent with the auditory system suppressing irrelevant and expected auditory information while (presumably) attending to visual streams [24,25,109].

Nevertheless, auditory stimuli may capture attention in a different manner depending on how easy they are to detect, i.e., saliency based (bottom-up processes) [110]. For example, the pitch conveyed in the fundamental frequency, F0, here in terms of the voice pitch, is a highly salient cue that plays an important role in the perceptual segregation of speech sources [111–113]. When speech is noise vocoded, saliency carried by the envelope periodicity of the speech is diminished [114–117]. In the context of our passive conditions, then, it is possible that noise-vocoded speech was not salient (or distracting) enough to elicit a reduction in cochlear gain sufficient to suppress irrelevant auditory information when attention was presumably focused elsewhere (i.e., on watching a silent film).

Interestingly, activation of the MOC reflex was observed for natural speech—further evidence that activation is not limited to tones and broadband noise [118–120]—and did not depend on whether participants were required to attend in a lexical decision task. This is consistent with natural speech being particularly salient as an ethologically relevant and nondegraded stimulus, as well as the low attentional load required when passively watching a film permitting continued monitoring of unattended speech [121].

To explain the variable reduction in cochlear gain between noise-vocoded and noise-masked speech across active and passive conditions (Fig 1E), top-down and bottom-up mechanisms can be posited as candidates to modify the activity of the MOC reflex. Bottom-up mechanisms, such as increased reflexive activation of the MOC reflex by wideband stimuli with flat power spectra [43] may explain why CEOAEs were suppressed during passive listening of speech masked by SSN (Fig 1E). The weaker activation of the reflex observed for speech in BN during passive listening may arise from BN more poorly activating the MOC reflex than "stationary" noises such as white noise or SSN [122,123]. However, if only bottom-up mechanisms were involved, then noise-vocoded speech with relatively fewer channels (i.e., Voc8 stimuli) might have also been expected to activate the MOC reflex more effectively due to their more "noise-like" spectra. The lack of suppression of CEOAEs in the passive noise-vocoded conditions, as well as the stimulus-specific pattern of MOC reflex activity in active listening conditions (i.e., enhanced suppression of cochlear gain for noise-vocoded versus preservation of cochlea gain for masked speech), suggests that a perceptual, top-down categorisation of speech sounds is necessary to engage or disengage appropriately the MOC reflex. Top-down mechanisms may include direct descending control of MOC fibre activity (either excitation or inhibition) or descending modulation of their sound-driven reflexive activity [22–24].

## Descending control of the MOC reflex for speech stimuli is likely bilateral

A central premise of our study, and of those exploring the effects of attention on the MOC reflex, is that OAEs recorded in one ear can provide a direct measure of top-down modulation of cochlear gain in the opposite ear. However, it has also been suggested that activation of the MOC reflex may differ between ears to expand interaural cues associated with sound localisation (while also enhancing amplitude modulations and suppressing noise in the ear with the better acoustic SNR) [124,125]. This process could be independently modulated by the largely ipsilateral corticofugal pathways evident anatomically (see [23] for review). Had the activation of the MOC reflex been independently controlled at either ear—for example, to suppress irrelevant clicks in one ear while preserving cochlear gain in the ear stimulated by speech—we would have expected similar suppression of CEOAEs across active and passive conditions for all speech manipulations—since click stimuli were always irrelevant to the task. Instead, however, suppression of CEOAEs—a biomarker for activation of the MOC reflex—was both stimulus and task dependent, reducing the likelihood that dichotic presentation of sounds engaged top-down modulation of cochlear gain differentially at either ear.

Anatomical evidence of purely ipsilateral corticofugal pathways ignores the possibility that, even when presented monaurally, descending control of the MOC reflex for speech stimuli may actually be bilateral. Unlike pure tones, speech activates both left and right auditory cortices even when presented monaurally [126]. In addition, cortical gating of the MOC reflex in humans does not appear restricted to direct, ipsilateral descending processes that impact cortical gain control in the opposite ear [127]. Rather, cortical gating of the MOC reflex likely incorporates polysynaptic, decussating processes that influence/modulate cochlear gain in both ears.

## Stimulus-specific enhancement of the neural representation of the speech envelope explains the pattern of CEOAE suppression

Beyond enhancing spatial listening [124] and protecting from noise exposure [128–130], the function most commonly attributed to activation of the MOC reflex is "unmasking" of transient acoustic signals in the presence of background noise [19,118,124,131]. By increasing the dynamic range of mechanical [16,17] and neural [18,19] responses to amplitude-modulated (AM) components in the cochleae [15,87] (see [28] for review), suppression of cochlear gain by the MOC reflex might preferentially favour the neural representation of syllables and phonemes in speech over masking noises.

While models of the auditory periphery that include efferent feedback typically demonstrate improved word recognition for a range of masking noises [pink noise [58–60], SSN [61] and BN [59]], experimental studies only sporadically report positive correlations between increased activation of the MOC reflex and improved speech-in-noise perception [29,33,34], with some even reporting negative correlations [27,35,132] or no effect at all [36,37]. Although this variability has generally been attributed to methodological differences in the measurement of OAEs [29,38], the majority of these studies have assessed contralateral activation of the MOC reflex and performance in a speech-in-noise task in separate sessions. As a result, the possibility that the MOC reflex's automatic activation is modulated by top-down processes to maximise its functional relevance in auditory tasks has not been fully challenged.

Here, we simultaneously employed degraded speech tokens as both contralateral activators of the MOC reflex and as targets in a lexical decision task and, therefore, were able to ascertain directly the involvement of the MOC reflex in both active and passive tasks. The stimulus- and task-dependent suppression of cochlear gain we observed suggests that automatic activation of the MOC reflex is indeed gated on or off by top-down modulation, with the direction of this

modulation dependent on whether or not the reflex functionally benefits performance in either task, i.e., facilitating a lexical decision in an active listening task or ignoring the auditory stimulus in a passive one.

Our modelling data support this conclusion by accounting for the stimulus-dependent suppression of CEOAEs through enhancement of the neural representation of the speech envelope in the AN. The apparent benefit of suppressing cochlear gain in response to envelopes of noise-vocoded words, compared to a disbenefit for words masked by background noise, is consistent with noise-vocoded words retaining relatively strong envelope modulations and that these modulations are extracted effectively through expansion of the dynamic range as cochlear gain is reduced. For both noise maskers, however, any improvement to envelope coding due to dynamic expansion of the mechanical (and neural) range applies to both speech and masker since these signals overlap spectrotemporally. Thus, the reduction of cochlear gain results in a poorer representation of the speech envelope.

### Dependence of MOC reflex on SNR

The results of our simulations are based on average changes in the neural correlation coefficient with efferent feedback calculated for 100 words. For individual words, however, the effect of suppressing cochlear gain was highly dependent on both the word itself and the SNR at which it was presented (Fig 2E, S3B Fig). If top-down modulation gates activation of the MOC reflex, it must account for the statistical likelihood that suppression of cochlear gain can improve the neural coding of the stimulus envelope throughout the task.

A potential criterion for predicting whether activation of the MOC reflex can improve speech intelligibility is the extent to which target speech is "glimpsed" in spectrotemporal regions least affected by a background noise [133–135]. If computing SNR of the speech envelope in short-time windows underpins speech intelligibility—as proposed by several models [51,68,69]—then this could be a suitable metric by which top-down modulation of the MOC reflex is adjusted. This metric could explain why previous studies of CEOAE suppression, which share identical paradigms aside from the stimuli with different spectrotemporal content (i.e., consonant–vowel pairs), can generate opposing correlations between discrimination in noise and the strength of the MOC reflex [27,34]. Additionally, presenting words at different SNRs should impact any benefit on speech intelligibility of activating the MOC reflex and can explain reported correlations between the strength of the MOC reflex and task performance at a range of SNRs [30,38].

In our model, switching from lowest to intermediate stimulus SNRs for both noise maskers led to an increase in the average $\Delta\rho$ENV when efferent feedback was applied (S3B Fig). This contributed to a small improvement in envelope encoding for +10 dB SNR speech in BN. While the absence of CEOAE suppression experimentally in the active task for BN at +10 dB SNR did not reflect the modelling results (Fig 1F), only a slight majority (59/100) of $\Delta\rho$ENV values were positive for speech in BN at +10 dB SNR, and this may have changed were different words selected. The modelling results for the intermediate SNR in the SSN condition (+8 dB SNR) were, however, consistent with the lack of CEOAE suppression observed experimentally in the active task (Fig 1F). Where an active MOC reflex does not functionally benefit the neural representation of the speech envelope (e.g., at negative SNRs), it remains possible that it is activated in another capacity, for example, to prevent damage by prolonged exposure to loud sounds [128–130].

### Reconciling stimulus-specific effects of the MOC reflex with previous studies using MAP model and efferent feedback

Resolving the differences between our stimulus-dependent observations in the Matlab Auditory Periphery and Brainstem (MAP_BS) model and those of previous MAP models

incorporating efferent feedback and an automatic speech recogniser (ASR) (i.e., consistent improvements to speech-in-noise recognition, irrespective of the type of noise masker) requires understanding the models' different outputs and analyses. While MAP and MAP_BS models share many similarities [58–60,62–64,136,137], they differ in their AN fibre outputs. The MAP model generates estimated spike probabilities across populations of simulated AN fibres [137], whereas the spike train output of the MAP_BS model is more stochastic in nature and incorporates the effects of internal noise in individual AN fibres [136,138].

Here, we took advantage of the MAP_BS's output to present stimuli of opposing polarity and compare coincident events in the resulting stochastic spike trains. Consequently, by extracting and quantifying these envelope-sensitive components with *sumcor* analysis (Fig 2C and 2E), we could identify how envelope encoding was affected by the suppression of cochlear gain for differently degraded words [73–76] and provide a coherent rationale for our experimental observations (Fig 2E and 2F). The ASR, on the other hand, has been previously used as a proxy for human hearing, predicting the intelligibility of test digit sequences from their AN spike probabilities using a hidden Markov model [58–60,62,139]. While both stimulus envelope and TFS cues are available to the ASR during digit classification, their respective weighting is not transparent as the ASR exploits any available cue to correctly identify the digit sequence. Therefore, a small enhancement of TFS encoding by the MOC reflex (as observed in our simulations; S5 Fig) could potentially outweigh any disbenefits to envelope encoding. This would result in improved digit identification by the ASR without explaining experimental CEOAE results for masked speech, such as we present here.

Even if envelope encoding had been weighted strongly by the ASR in previous studies, its representation of the neural envelope was likely very different to the *sumcor* analysis. This is particularly evident when comparing the effects of MOC reflex on "natural," clean speech in this study and that of Brown and colleagues who also used fixed attenuation of the MOC reflex [58]. Whereas we observe improved encoding of the stimulus envelope in the 4 to 8 Hz range of modulation frequencies for natural sentences (S1B Fig), the ASR's recognition of digits in silence was reduced when the MOC reflex was active (Fig 7A in [58]). This decrease in classification accuracy arises from the ASR interpreting the effects of the MOC reflex as an overall reduction in AN output (due to suppression of the cochlea gain) rather than the generation of sparser yet more precise neural envelopes as suggested by *sumcor* analysis and calculation of ρENV (Fig 2E and 2F). In addition, were we to present noise-vocoded speech—a single stream like "natural," clean speech—to the ASR, its AN output would also appear reduced with the MOC reflex, impacting the ASR's word recognition and further highlighting its inability to explain our experimental CEOAE data.

## Considerations when modelling the MOC reflex

Many aspects of the MOC reflex and its circuitry remain poorly understood. For example, while we represent the active MOC reflex here as a fixed attenuation of cochlear gain (allowing a more controlled examination of the reflex's effects on envelope encoding), it is in fact a dynamic process with multiple time courses (spanning fifty milliseconds to tens of seconds) whose purposes remain unknown [45,140–142]. Previous MAP models incorporating dynamic efferent feedback have gravitated towards slow time constants to optimise speech recognition by their ASR [59,60,62]. However, since digit recognition by the ASR always improved with the MOC reflex, whatever its chosen time constant [60], we are confident that introducing a dynamic MOC reflex would not alter the stimulus dependence of any effects we observed in our model.

An added complexity of modelling MOC reflex activity is considering how top-down gating modulates the reflex and over what timescale. It remains unknown whether corticofugal inputs

to the MOC reflex produce tonic activation/inhibition or only modulate the reflex's strength as has been shown for awake versus anaesthetised animals [20,23,143–145]. Given anatomical evidence that MOC neurons sparsely innervate broad regions of the cochlea [146,147], it is possible that, through a combination of tonic top-down modulation and poor frequency tuning of efferent innervation, the fixed attenuation we have implemented across all frequency channels may closely reflect actual recruitment of the MOC reflex under active listening conditions. Finally, our model does not include neural adaptation to stimulus statistics as a potential contributing factor to speech-in-noise discrimination [86,87]. We therefore cannot discount its involvement in the lexical decision task nor show that it is sufficient for robust recognition of degraded speech without the MOC reflex as has been previously suggested [86,87,148].

## Higher auditory centre activity supports coexistence of multiple strategies to achieve similar levels of performance

The impact of modulating the MOC reflex was observed in the activity of the auditory midbrain and cortex. Increased midbrain activity for active noise-masked conditions was consistent with changes in magnitudes of ABRs previously reported during unattended versus attended listening to speech [149] or clicks [150,151]. This highlights the potential for subcortical levels either to enhance attended signals or filter out distracting auditory information. At the cortical level, recorded potentials were larger for all attended, compared to ignored, speech (S6A–S6C Fig), consistent with previous reports [152,153]. However, later cortical components were larger for masked, compared to noise-vocoded, speech while attending to the most extreme manipulations. Late cortical components have been associated with the evaluation and classification of complex stimuli [104] as well as the degree of mental allocation during a task [103]. Therefore, differing cortical activity likely reflects greater reliance on, or at least increased contribution from, context-dependent processes for speech masked by noise than for noise-vocoded speech.

Together, differences in physiological measurements from higher auditory centres and the auditory periphery highlight the possibility of diverging pathways to process noise-vocoded and masked speech. Evidence for systemic differences in processing single, degraded streams of speech, compared to masked speech, has been reported in the autonomic nervous system [107]: where, despite maintaining similar task difficulty across conditions, masked speech elicits stronger physiological reactions than single unmasked streams. Here, we propose that 2 strategies enable isoperformance to be maintained even when stimuli are categorically different. The processing of single and intrinsically degraded streams selectively recruits auditory efferent pathways from the AC to the inner ear which, in turn, "denoises" the representation of the stimuli in the periphery (Fig 3B). By contrast, multiple streams, represented by speech in BN or SSN, appear to rely much more on higher auditory centres such as midbrain and AC for the extraction of foreground, relevant signals (Fig 3B). Given evidence of denoised auditory signals in the cortex [47,154–156], the extensive loops and chains of information between cortex, subcortical regions, and the auditory periphery in everyday listening environments [20,157–159] have not been acknowledged, nor has their candidacy as targets for hearing technologies.

## Implications for hearing-impaired listeners

Although normal-hearing listeners appear to benefit from an MOC reflex that modulates cochlear gain and is amenable to top-down, attentional control, it is important to note that users of CIs—for whom normal-hearing listeners processing noise-vocoded speech are often considered a proxy—have no access to the MOC reflex. CIs bypass the mechanical processes

of the inner ear, including the OHCs—which are nonfunctioning or absent completely in individuals with severe to profound hearing loss—to stimulate directly the AN fibres themselves. Efforts have been made to incorporate MOC-like properties into CI processes, providing expanded auditory spatial cues and "denoised" electrode output to improve listening in bilaterally implanted CI users [124,125], but the capacity to exploit efferent feedback to aid speech understanding in CI listeners is yet to be realised in any device.

Most recently, Lopez-Poveda and colleagues [125] highlighted the benefits of introducing an MOC strategy to CI users' understanding of speech in both quiet and noise. Consistent with this, our observed activation of MOC reflex for noise-vocoded speech supports the notion that enhancing the neural representation of acoustic envelopes is key to understanding intrinsically degraded speech for both normal-hearing listeners and CI users (Fig 1F). However, masked speech generates different outcomes between the 2 groups; the improved speech understanding of CI users using an MOC strategy may not be solely related to envelope expansion but may also derive from increased neural "unmasking" due to a reduction of CI stimulation by the MOC strategy [125]. In normal-hearing individuals, however, who showed no MOC reflex activation when listening to masked speech (Fig 1F), the "unmasking" rationale may not be as critical given the far larger dynamic ranges of their AN fibres compared to CI-using counterparts [160–162]. This suggests that any future implementation of MOC strategies in CIs might not necessarily reflect the fundamental role of the reflex in a healthy auditory system.

For other hearing-impaired listeners, aided or unaided, the contribution of MOC reflex feedback to speech processing is limited compared to normal-hearing listeners as, in most cases, their hearing loss comes from damage to the OHCs, which receive direct synaptic input from MOC fibres. In hearing loss, generally, the degradation or loss of peripheral mechanisms contributing to effective speech processing in complex listening environments may mean that listeners rely more heavily on attentional and other cortical-mediated processes, contributing to widely reported increases in listening effort required to achieve adequate levels of listening performance [108]. This increase in listening effort—likely manifesting over time—may not be reflected in performance in relatively short, laboratory-, or clinic-based assessments of hearing function.

## Materials and methods

This study was approved by the Human Research Ethics Committee of Macquarie University (ref: 5201500235) and was performed according to the Australian Code for the Responsible Conduct of Research. Each participant signed a written informed consent form and was provided with a small financial remuneration for their time.

### Hearing assessment

A total of 56 participants (36 females, aged between 18 and 35 [mean: 24 ± 7 years old]) were recruited in this study; however, not all subjects completed every experimental measurement (S2 Table). All subjects included in this study had normal pure tone thresholds (<20 dB HL); normal middle ear function (standard 226 Hz tympanometry); and normal OHC function (assessed with distortion product otoacoustic emissions (DPOAEs) between 0.5 and 10 kHz).

### MEMR assessment and stimuli calibration

Controlling the stimulus level is a critical step when recording any type of OAE due to the potential activation of the middle ear muscle reflex (MEMR). High-intensity sounds can evoke contractions of both the stapedius and the tensor tympani muscles, causing the ossicular chain

to stiffen and the impedance of middle ear sound transmission to increase. As a result, retrograde middle ear transmission of OAE magnitude can be reduced due to MEMR and not MOC reflex activation [163]. It has been shown that even sounds 10 to 15 dB SPL below the clinical MEMR threshold can cause contractions of the middle ear muscles [164–166]. Therefore, a modified version of the clinical protocol (Titan, Interacoustics, Middelfart, Denmark) was used for threshold estimation of the MEMR. Due to the broadband nature of our experimental stimuli (i.e., clicks and speech), instead of tones (typical clinical paradigm), wideband (0.25 to 8 kHz) stimuli were used as activators of both the contralateral and ipsilateral MEMR. MEMR activations were monitored in a modified range (60 to 80 dB HL) with a 5-dB step size and a very sensitive threshold criterion (0.02 ml). All participants had thresholds >75 dB HL. Therefore, presentation level for all natural, noise-vocoded, and speech-in-noise tokens was set at 75 dBA (root–mean–square normalised) and click stimulus at 75 dB p-p. According to ANSI S3.6–1996 standards for the conversion of dB SPL to dB HL, a minimum of 10 dB SPL difference across frequencies was kept between our participants MEMR thresholds (>75 dB HL) and stimulus levels. Therefore, no significant impact of MEMR was expected in our experimental paradigm.

### Experimental protocol

Participants were seated comfortably inside an electrically shielded, sound-proof booth (ISO 8253–1:2010) while wearing an EEG cap (Neuroscan 64 channels, SynAmps2 amplifier, Compumedics, Melbourne, Australia). Two attentional conditions (passive and active) were counterbalanced across participants. In the passive listening condition, subjects were asked to ignore the auditory stimuli and to watch a non-subtitled, stop motion movie. To ensure participants' attention during this condition, they were monitored with a video camera and were asked questions at the end of this session (e.g., What happened in the movie? How many characters were present?). The aim of a passive or an auditory-ignoring condition is to shift attentional resources away from the auditory scene and towards the visual scene. During active listening, participants performed an auditory lexical decision task, where they were asked to press a keyboard's space key each time they heard a non-word in strings of 300 speech tokens. D prime (d') was used as a measure of accuracy and calculated as: Z(correct responses)–Z(false alarm) (i.e., Z(correct responses)) = NORMSINV(correct responses). Simultaneous to the presentation of word/non-word in one ear, CEOAEs were recorded continuously in the contralateral ear (Fig 1A). The ear receiving either the clicks or speech stimuli was randomised across participants.

### Speech stimuli

A total of 423 word items were acquired from Australian-English-adapted versions of monosyllabic consonant–nucleus–consonant (CNC) word lists and were spoken by a female, native Australian-English speaker. The duration of words ranged between 420 and 650 ms. Moreover, 328 monosyllabic CNC non-word tokens were selected from the Australian Research Council nonword database. Speech stimuli were delivered using ER-3C insert earphones (Etymotic Research, Elk Grove Village, Illinois, USA) and Presentation software (Neurobehavioral Systems, Berkeley, California, USA, version 18.1.03.31.15) at 44.1 kHz, 16 bits. All tokens were root–mean–square normalised, and the calibration system (sound level metre (B&K G4) and microphone IEC 60711 Ear Simulator RA 0045 563 (BS EN 60645–3:2007) (see CEOAEs acquisition and analysis section)) was set to 75 dB "A-Weighting", which matches the human auditory range.

Each experimental condition, a combination of attentional and stimulus manipulations (see below for details of speech manipulations), was tested using 200 words and 100 nonwords (randomly selected from the speech corpus). Speech tokens were counterbalanced in each condition based on the presence of stop and nonstop initial consonants: 100 stop/nonstop

consonant words; 50 stop/nonstop consonants words with a maximum of 3 repeats per participant allowed. Each experimental condition had a duration of 12 minutes (Fig 1A), and participants could take short breaks between them if needed. The order of the experimental conditions was always randomised to prevent presentation order bias or training effects.

### Noise-vocoded speech

A total of 27 native speakers of Australian-English (17 females: 25 right-handed and 2 left handed) were recruited, aged between 18 and 35 (mean: 23 ± 5 years old). Based on the noise vocoding method and behavioural results of Shannon and colleagues [42], 3 noise-vocoded conditions (16, 12, and 8 channels: Voc16, Voc12, and Voc8, respectively) were tested to represent 3 degrees of speech intelligibility (i.e., task difficulty). Four stimulus conditions were assessed in both active and passive listening conditions: Stimulus condition 1: natural speech; Stimulus condition 2: Voc16; Stimulus condition 3: Voc12; and Stimulus condition 4: Voc8. Each experimental condition lasted 12 minutes (Fig 1A). The total of 8 experimental conditions (i.e., an active and passive condition for each of natural, Voc16, Voc12, and Voc8) had a 2.6 hours duration (including hearing assessment and EEG cap setup).

### Speech in BN

A total of 29 native speakers of Australian-English (19 females: 28 right handed, 1 left handed) were recruited, aged between 20 and 35 (mean: 26 ± 9 years old). The BN, used here, consisted of 4 females and 4 male talkers and was filtered to match the long-term average spectrum of the speech corpus (S7 Fig). Random segments from a 60-second BN recording were temporally matched to the speech tokens with no ramps applied to the stimuli (Fig 1C). Three stimulus conditions were presented in the active and passive listening conditions: Stimulus condition 1: natural speech; Stimulus condition 2: speech in BN at +10 dB SNR (BN10); and Stimulus condition 3: speech in BN at +5 dB SNR (BN5).

### Speech in SSN

The SSN was generated to match the long-term average spectrum long-term average of the speech corpus (S7 Fig). Random segments from a 60-second SSN were selected to temporally match the speech tokens (Fig 1C); no ramps were applied to the stimuli. Both BN and SSN manipulations were presented in the same session; therefore, Stimulus condition 1 was the same for both manipulations, Stimulus condition 2: speech in SSN at +8 dB SNR (SSN8) and Stimulus condition 3: speech in SSN at +3 dB SNR (SSN3). BN and SSN were combined into a unique session of 3 hours (including hearing assessment and EEG cap setup). The 29 subjects experienced a total of 10 experimental conditions, each of 12-minute duration (i.e., an active and passive condition for each of natural, BN10, BN5, SSN8, and SSN3).

### CEOAEs acquisition and analysis

Nonfiltered click stimuli, with a positive polarity and 83-μs duration were digitally generated using RecordAppX (Advanced Medical Diagnostic Systems, Oxford, Mississippi, USA) software. The presentation rate was 32 Hz in all conditions, which contributed to minimise ipsilateral MOC reflex activation [167]. Ipsilateral MOC reflex activation was otherwise constant across participants and experimental manipulations by maintaining a fixed click rate.

Both the generation of clicks and OAE recordings were controlled via an RME UCX soundcard (RME, Haimhausen, Germany) and delivered/collected to and from the ear canal through an Etymotic ER-10B probe connected to ER-2 insert earphones with the microphone pre-

amplifier gain set at 20 dB. Calibration of clicks was performed using a sound level metre (B&K G4) and microphone IEC 60711 Ear Simulator RA 0045 (BS EN 60645–3:2007). This setup was also used to calibrate the speech stimuli. In addition, clicks were calibrated in-ear using forward equivalent pressure level (FPL), ensuring accurate stimulus levels [168,169]. The OAE's probe was repositioned, recalibrated, and the block restarted if participants moved or touched it.

CEOAE data were analysed offline using custom Matlab scripts (available upon request). The averaged RMS magnitudes of CEOAE signals (Fig 1B) were analysed between 1 and 2 kHz given maximal MOC effects in this frequency band [46,170]. The energy in the 1 to 2 kHz CEOAE band does not necessarily originate solely from the equivalent tonotopic region in the cochlea, especially at high level intensities (such as the 75 dB p-p used here), where significant energy from nonlinear distortions distant to the 1 to 2 kHz region will likely contribute [171–173]. Given the broadband nature of the click stimuli and the sparse but nonfrequency specific nature of MOC innervation of the cochleae [146,147], the MOC reflex will be acting along the length of the cochleae when suppression is observed in the 1 to 2 kHz band. We therefore consider suppression of the cochlear gain in the 1 to 2 kHz band as a suitable and consistent marker for MOC reflex activity across the entire cochlea.

Only binned data for averaged CEOAEs displaying an SNR $\geq$ 6 dB (shown to reduce intra- and interindividual variability [29,170]) and with > 80% of epochs retained (i.e., had RMS levels within the 2 standard deviations limit) were selected as valid signals for further analysis; see example individual data (S8 Fig). Although 2 minutes of baseline CEOAE were recorded at the beginning and end of each block, in the absence of speech tokens (Fig 1A), only the first minute was used as baseline, due to low SNR and high number of artefacts (participants swallowing and jaw movements) in the last minute of CEOAE recordings. As no significant differences were observed between CEOAE baseline magnitudes within participants ($p > 0.05$) across experimental conditions, all baselines were pooled within participants. This allowed for an increase in both SNR and reliability of the individual CEOAE recordings. After baseline recording, CEOAEs were continuously obtained for 10 minutes during the contralateral presentation of the speech tokens (Fig 1A). The suppression of CEOAE magnitude (dB SPL) relative to the baseline was calculated as follows and reported as means and SEM:

$$\text{CEOAE suppression} = \text{CEOAE}_{\text{speech presentation (average across minutes)}} - \text{CEOAE}_{\text{baseline (first 60 s)}}.$$

## EEG: Event Related Potentials (ERPs)

EEG measurements and the CEOAE setup were synchronised using a Stimtracker (https://cedrus.com/) (Fig 1A and 1C). EEG data were acquired according to the 10 to 20 system (internationally standardised scalp electrode placement [174]). Impedance levels were kept below 5 kΩ for all electrodes. Signals were sampled at a rate of 20 kHz in the AC mode with a gain of 20000 and an accuracy of 0.15 nV/least significant bit (LSB). Early and late ERP components were analysed offline using fieldtrip-based scripts. Data were rereferenced to the average of mastoid electrodes. Trials started 200 ms before and ended 1.2 seconds after speech onset. Components visually identified as eye blinks and horizontal eye movement were excluded from the data as well as trials with amplitude >75 μV. The accepted trials (60% to 80% per condition) were band-pass filtered between 0.5 and 30 Hz with transition band roll-off of 12 dB/octave. Trials were baseline-corrected using the mean amplitude between −200 and 0 ms before speech onset. Baseline-corrected trials were averaged to obtain ERP waveforms (Fig 1C). Analysis windows centred on the grand average ERP component maximums were selected: P1 (100 to 110 ms) and N1(145 to 155 ms); P2 (235 to 265 ms), N400 (575 to 605 ms); and LPP (945 to 975 ms) (Fig 1C). Mean amplitude for each component within the analysis window was calculated for each participant and experimental condition.

### EEG: Auditory Brainstem Responses (ABRs)

ABR signals were extracted from central electrodes (FZ, FCZ, and CZ). Moreover, 16-ms duration ABR analysis windows (2 ms prior and 14 ms after click onset-stimulus artefact between 0 and 1 ms) were selected (Fig 1B). A total of 19,200 trials (click rate of 32 Hz across 10 minutes per condition) were band-pass filtered between 200 and 3,000 Hz. Averaged ABR waveforms were obtained using a weighted-averaging method [175,176]. Amplitude of waves III (peak amplitude at 4 ms across) and V (peak amplitude at 6 ms) (Fig 1B) were visually determined by the first author and 2 lab members (nonauthors) for each subject across blocks and conditions when appropriate (wave amplitudes above the residual noise, therefore a positive SNR, Fig 1B). Due to stimulus level restrictions ($< = 75$ p-p dB SPL to avoid MEMR activation), wave I could not be extracted from the EEG residual noise.

### Statistical analysis

Sample size estimation was computed according to the statistical test employed by using G*Power (Effect size f = 0.4; $\alpha$ err prob = 0.05; Power (1-$\beta$ err prob) = 0.8). All variables were tested for normality (Shapiro–Wilk test); outlier residual values preventing normal distribution were removed from the data set (S2 Table). One-way ANOVA for the behavioural and ERPs data and rANOVA for CEOAEs and ABRs data and $t$ tests (alpha = 0.05, with Bonferroni corrections for multiple comparisons) were performed. One-way ANOVAs had stimulus type (i.e., Natural, Voc16, BN10) as factors, whereas rANOVA had both attentional conditions (active and passive) and stimulus type as factors. The interaction between factors was also explored. Effect sizes were calculated for all statistical analysis (Eta-squared ($^2$) for ANOVAs, and Cohen's d was reported for all $t$ tests) [177].

### AN simulations

The MAP_BS [64,65] computational model was used to simulate AN responses with and without efferent feedback (MOC reflex) in thirty frequency channels, logarithmically spaced between 0.1 kHz and 4.5 kHz. Similarly to previous versions of the model [58–60,62,63,139], MAP_BS uses a dual-resonance nonlinear (DRNL) filter bank to translate the input of the "outer/middle ear" stage into "basilar membrane velocity" in each frequency channel (Fig 2A). Both linear and nonlinear paths of the DRNL consist of a sequence of band-pass (Gammatone) and low-pass (Butterworth) filters; however, the nonlinear path also includes a compressive nonlinearity that acts when the stimulus exceeds a threshold level. For the current simulations, MAP_BS was run in the "AN only" mode at 100 kHz, i.e., no brainstem neurons were simulated, and stochastic spike trains were generated as the AN output (as opposed to spiking probability of previous MAP models) [58–60,62–64,136,139]. A total of 400 AN fibres (200/ear) were simulated for each natural and degraded word token. Although we simulated both low-threshold, HSR, and high-threshold, low-spontaneous rate AN fibres (modelled by setting the calcium clearance time constant to $2.4 \times 10^{-4}$ s for HSR and $0.8 \times 10^{-4}$ s for LSR), the latter were considered as the main fibre type given their suggested importance for speech in noise at high stimulus intensities [70–72].

### Implementation of efferent feedback in simulations

MOC attenuation of the cochlear gain in the MAP_BS model was implemented at the first stage of the DRNL filterbank's nonlinear path. Given the purpose of the modelling was to observe the qualitative effect that suppression of cochlear gain had on the neural envelope of differentially degraded word tokens (as opposed to matching behavioural data quantitively or

optimising the effect quantitively as in previous studies [58–60,62,139]), we chose to represent the active MOC reflex condition as a fixed, nontemporally varying attenuation of the cochlear gain. This not only imposed a suppression of the cochlear gain that was consistent across all word tokens variants, hence avoiding any stimulus-dependent differences in the time course or mean activation of the MOC reflex for noise-vocoded and masked versions of the same word, but also accelerated simulations as we could present degraded and natural word tokens to the model individually without worrying about the MOC reflex's initial strength at stimulus onset. Values of 10/15 dB attenuation were applied for the fixed efferent feedback given their effective action on stimuli of similar nature and intensity in previous versions of the model [58,59]. The MEMR was not active in the model as the "ARatt" parameter was set to 0.

## Word presentation

A total of 100 words (50 stop/nonstop consonant words) were chosen at random from the speech corpus and were degraded using the most demanding speech manipulations (Voc8, BN5, and SSN3; Fig 1D). Normal, Test+ and polarity-inverted, Test- versions of each manipulation were presented to the MAP_BS model at 75 dB SPL both with and without efferent feedback (Fig 2A). Natural words (both normal, Nat+, and polarity-inverted, Nat-) were also presented to the MAP_BS model with and without efferent feedback; however, the AN output with active efferent feedback was selected as the main reference condition to compare against neural responses to degraded speech tokens (for exception, see S2 Fig).

## Shuffled autocorrelogram analysis

Comparative analysis of AN coding of AM envelope between Voc8/BN5/SSN3 conditions and the reference natural condition (with the MOC reflex) was performed using shuffled auto- and cross- correlograms (SACs and SCCs, respectively) [50,73,74]. Normalised all-order histograms were calculated using the spike trains of 400 high-threshold AN fibres with a coincidence window of 50μs and a delay window ± 25 ms centred on zero [73]. No correction for triangular shape was required given brevity of delay window relative to stimulus length (between 420 and 650 ms) [74,77]. A neural cross-correlation coefficient, $\rho ENV$, quantifying AM envelope encoding similarity between conditions was generated as follows [50,73]:

$$\rho ENV = \frac{(sumcor_{Test\,Nat} - 1)}{\sqrt{(sumcor_{Test} - 1) \times (sumcor_{Nat} - 1)}},$$

where $sumcor_{Nat}$ (natural word reference) and $sumcor_{Test}$ (Voc/BN/SSN conditions) are the averages of SACs (Normalised all-order histograms for Nat+ versus Nat+/Test+ versus Test + for $sumcor_{Nat}$ /$sumcor_{Test}$, respectively) and cross-polarity histograms (Normalised all-order histograms for Nat+ versus Nat-/Test+ versus Test- for $sumcor_{Nat}$ /$sumcor_{Test}$, respectively). $Sumcor_{Test\,Nat}$ is the average of the SCC (Average of normalised all-order histograms for Nat + versus Test+ and Nat- versus Test-) and the cross-polarity correlogram (Average of normalised all-order histograms for Nat- versus Test+ and Nat+ versus Test-) between natural and Voc8/BN5/SSN3 conditions. All high-frequency oscillations (> characteristic frequency of AN fibre), associated with fine-structure leakage, were removed from $sumcor$s [73,76]. $\rho ENV$ values ranged from 0 to 1 where 0 represents completely dissimilar spike trains and 1 represents identical spike patterns [50,73]. The neural cross-correlation coefficient, $\rho TFS$, was also calculated to quantify the similarity of 2 conditions' TFSs as follows [50,73]:

$$\rho TFS = \frac{(diffcor_{Test\,Nat})}{\sqrt{(diffcor_{Test}) \times (diffcor_{Nat})}},$$

where diffcor$_{Nat}$ (natural word reference) and diffcor$_{Test}$ (Voc/BN/SSN conditions) are the difference between SACs (Normalised all-order histograms for Nat+ versus Nat+/Test+ versus Test+ for diffcor$_{Nat}$ /diffcor$_{Test}$, respectively) and cross-polarity histograms (Normalised all-order histograms for Nat+ versus Nat-/Test+ versus Test- for diffcor$_{Nat}$ /diffcor$_{Test}$, respectively). Difcor$_{Test\ Nat}$ is the difference between the SCC (Average of normalised all-order histograms for Nat+ versus Test+ and Nat- versus Test-) and the cross-polarity correlogram (Average of normalised all-order histograms for Nat- versus Test+ and Nat+ versus Test-) between natural and Voc8/BN5/SSN3 conditions. $\rho TFS$ was calculated only for masked speech stimuli (BN5 and SSN3) given that noise vocoding scrambles the TFS cues and, therefore, would limit the utility of any comparison of TFS encoding for noise-vocoded and natural speech [50,178].

## Analysis of modelling and statistics

Percentage changes in $\rho ENV$ due to efferent feedback inclusion in MAP_BS were calculated for each test frequency and Voc8/BN5/SSN3 condition as follows:

$$\Delta \rho \mathrm{ENV}_{freq} = \left( \frac{\rho ENV_{eff} - \rho ENV_{no\ eff}}{\rho ENV_{no\ eff}} \right) \times 100,$$

where $\Delta \rho ENV_{freq}$ is the percentage change in $\rho ENV$ at a test-frequency for a manipulated word. $\rho ENV_{no\ eff}$ and $\rho ENV_{eff}$ are measures of $\rho ENV$ with and without efferent feedback enabled, respectively. An average $\Delta \rho ENV_{freq}$ was calculated across test-frequencies for each word and manipulation. Similar calculations were performed for $\Delta \rho TFS_{freq}$ by replacing $\rho ENV$ in the equation with $\rho TFS$. Data are reported as means and SEM. A one-sample Wilcoxon signed rank test (not normally distributed data) was performed to confirm whether average $\Delta \rho ENV_{freq}$ for all words differed from zero for each speech manipulation. Paired Wilcoxon signed rank tests were performed between experimental conditions. Wilcoxon effect size (r) was calculated for all statistical tests.

## Supporting information

**S1 Table. Planned *t* test comparisons between CEOAEs baseline measures and CEOAEs magnitude obtained during the presentation of noise-vocoded speech and masked speech.** CEOAE, click-evoked otoacoustic emission. (XLSX)

**S2 Table. Subjects removed for CEOAEs suppression and ABR analysis.** ABR, auditory brainstem response; CEOAE, click-evoked otoacoustic emission. (XLSX)

**S1 Fig. Efferent feedback improves envelope encoding for naturally spoken sentences. (A)** Shuffled-Correlogram *Sumcors* (upper panel) were calculated for the naturally spoken sentence, "the steady drip is worse than the drenching rain" (s86; The MAVA corpus, [179]), using LSR AN fibre output in the 2.334 kHz channel with (red line) and without (black line) efferent feedback (MOC reflex). A longer, 1-second delay window was used compared to the single word presentation; in addition, inverted triangular compensation was implemented to compensate for large delays relative to signal length [74,77]. The envelope power spectral density (lower panel) was computed both with (red line) and without (black line) efferent feedback by computing Fourier transforms of the above *Sumcors* with a <1 Hz spectral resolution. Efferent feedback was conducive to larger envelope responses, especially at low modulation

frequencies associated with words and syllables. **(B)** Envelope power spectra computed with and without MOC reflex in the 4 to 8 Hz modulation range for 6 sentences (s7, s26, s37, s42, s86, and s164; The MAVA corpus, [179]). In all instances, adding MOC reflex improved envelope encoding across most modulation frequencies. The underlying data can be found in https://doi.org/10.5061/dryad.3ffbg79fw. AN, auditory nerve; LSR, low spontaneous rate; MOC, medial olivocochlear.
(EPS)

**S2 Fig. Using a different control template condition (Nat$_{ANonly}$) to calculate $\rho$ENV$_{AN}$ does not alter stimulus-specific changes to envelope encoding when (15-dB attenuation) efferent feedback is added to LSR fibres. (A)** $\Delta\rho$ENVs for 100 words [in their 3 degraded forms and for high- and low-frequency bands [<1.5 kHz (left, A) and >1.5 kHz (right, A)] were calculated using AN responses to "Natural" speech (i.e., in quiet) presented without the MOC reflex as the control template to compute values of $\rho$ENV$_{AN}$, i.e., Nat$_{ANonly}$ versus Degraded$_{ANonly}$. $\Delta\rho$ENVs for all manipulations in both low and high-frequency bands followed the same stimulus-dependent trends as in Fig 2. (mean $\Delta\rho$ENV for Voc8 for freqs <1.5 kHz = +-3.69 ± 0.30%, [Z(99) = 5.68, $p <$ 0.001, r = 0.57]; mean $\Delta\rho$ENV for Voc8 for freqs >1.5 kHz = + 8.08 ± 0.36%, [Z (99) = 8.6, $p <$ 0.0001, r = 0.87]; mean $\Delta\rho$ENV for BN5 for freqs <1.5 kHz = −9.20 ± 0.66, [Z(99) = −8.68, $p <$ 0.0001, r = 0.87]; mean $\Delta\rho$ENV for BN5 for freqs >1.5 kHz = −3.24 ± 0.30%, [Z(99) = −7.84, $p <$ 0.001, r = 0.57]; mean $\Delta\rho$ENV for SSN3 for freqs <1.5 kHz = −9.09 ± 0.70, [Z (99) = −8.68, $p <$ 0.0001, r = 0.87]; mean $\Delta\rho$ENV for SSN3 >1.5 kHz = −5.60 ± 0.34, [Z (99) = −8.68, $p <$ 0.0001, r = 0.87]). $\Delta\rho$ENVs for Voc8 stimuli (pink circles, left, A) were exclusively positive in the high-frequency band with the largest benefits observed for noise-vocoded tokens with the lowest $\rho$ENV$_{AN}$ values, as observed in Fig 2E. In addition, the most negative $\Delta\rho$ENVs for BN5 and SSN3 stimuli were observed for the lowest values of $\rho$ENV$_{AN}$. **(B)** Comparing $\Delta\rho$ENVs calculated using Nat$_{ANonly}$ and Nat$_{AN+MOC}$ as a control template for $\rho$ENV$_{AN}$ at high frequencies (>1.5 kHz). The mean improvement in envelope encoding for Voc8 stimuli was larger after calculating $\rho$ENV$_{AN}$ with the new Nat$_{ANonly}$ control template ([Z(99) = −4.6, $p <$ 0.001, r = 0.47]) (left column, B). Similarly for masked stimuli (BN5 (middle, B) and SSN3 (right, B)), the new control template for $\rho$ENV$_{AN}$ led to an increase in the impairment to envelope encoding with the MOC reflex (BN5:[Z(99) = −6.50, $p <$ 0.001, r = 0.65]; SSN3:[Z(99) = −7.09, $p <$ 0.01, r = 0.71]) (middle and right columns, B). The underlying data can be found in https://doi.org/10.5061/dryad.3ffbg79fw. AN, auditory nerve; BN, babble noise; LSR, low spontaneous rate; MOC, medial olivocochlear; SSN, speech-shaped noise.
(EPS)

**S3 Fig. Comparing mean changes in $\Delta\rho$ENVs (in >1.5kHz frequency band) for control conditions. (A)** Using a smaller fixed attenuation for the active MOC reflex (10-dB attenuation) than in the main simulations (15-dB attenuation) reduced both positive mean $\Delta\rho$ENV for Voc8 and negative mean $\Delta\rho$ENVs for BN5/SSN3 (Voc8: [Z(99) = −7.94, $p <$ 0.001, r = 0.79]; BN5: [Z(99) = −5.53, $p <$ 0.001, r = 0.55]; SSN3: [Z(99) = −7.85, $p <$ 0.001, r = 0.78]). Nevertheless the benefits (Voc8) and disbenefits (BN5/SSN3) of adding the MOC reflex to envelope encoding remained for all 3 stimulus manipulations (Voc8: ([Z(99) = 6.756, $p <$ 0.0001, r = 0.79]; BN5: [Z(99) = −5.16, $p <$ 0.001, r = 0.52]; SSN3: [Z(99) = −8.44, $p <$ 0.0001, r = 0.84]). **(B)** Presenting degraded speech tokens with more channels for Voc stimuli, i.e., Voc16, generated signficantly larger $\Delta\rho$ENVs with a fixed 15-dB MOC reflex attenuation ([Z(99) = −3.66, $p <$ 0.001, r = 0.4]). Increasing the SNRs (10-dB SNR for BN and 8 dB SNR for SSN) significantly reduced $\Delta\rho$ENVs for speech-in-noise conditions when the same MOC reflex attenuation was implemented (BN: [Z(99) = −8.20, $p <$ 0.001, r = 0.82]; SSN: [Z

(99) = −8.67, $p < 0.001$, r = 0.87]). For BN10, the new mean $\Delta\rho$ENVs was in fact positive ([Z (99) = 2.025, $p = 0.043$, r = 0.2]). The underlying data can be found in https://doi.org/10.5061/dryad.3ffbg79fw. BN, babble noise; MOC, medial olivocochlear; SNR, signal-to-noise ratio; SSN, speech-shaped noise.
(EPS)

**S4 Fig. Percentage change in envelope encoding after introduction of (15-dB attenuation) efferent feedback to low-threshold, HSR AN fibres (in > 1.5kHz frequency band). (A and B)** $\Delta\rho$ENVs for 100 words (in their 3 degraded forms) were calculated as in Fig 2E; however, HSR AN fibre output for frequencies >1.5 kHz was used here. $\Delta\rho$ENVs for Voc8 words (pink circles, A) varied greatly (Max-Min $\Delta\rho$ENV for Voc8 > 1.5kHz = +32.84 to −0.43%) but the mean $\Delta\rho$ENV was significantly positive (mean $\Delta\rho$ENV for Voc8 >1.5 kHz = +12.06 ± 0.57%, [Z (99) = 8.68, $p < 0.0001$, r = 0.87]). Note that values of $\rho$ENV$_{AN}$ for Voc8 were smaller here compared to values for low SR (LSR) AN fibres (mean $\rho$ENV$_{AN-HSR}$ for Voc8 >1.5 kHz = 0.55 ± 0.01 versus mean $\rho$ENV$_{AN-LSR}$ for Voc8 >1.5 kHz = 0.64 ± 0.01). By contrast, the distributions of $\Delta\rho$ENVs for BN5 (light blue squares, A) and SSN3 (green diamonds, A) appeared more compact (Max-Min Range $\Delta\rho$ENV for BN5 words = +2.062,27 to −9.79%; Max-Min $\Delta\rho$ENV for SSN3 = +1.07 to −10.57%); however, as for LSR AN fibre results (Fig 2E and 2F), both mean $\Delta\rho$ENVs for HSR AN fibres were significantly negative overall (mean $\Delta\rho$ENV for BN5 = −3.47 ± 0.27, [Z(99) = −8.18, $p < 0.001$, r = 0.82]; mean $\Delta\rho$ENV for SSN3 = −4.36 ± 0.22, [Z(99) = −8.65, $p < 0.0001$, r = 0.87]). Progression of mean $\Delta\rho$ENVs (± SEM) for model data > 1.5kHz (*checkerboard bars*, right, B) mirrored that of active-task, CEOAE data (mean ± SEM) (solid colour bars, left, B). The underlying data can be found in https://doi.org/10.5061/dryad.3ffbg79fw. AN, auditory nerve; BN, babble noise; CEOAE, click-evoked otoacoustic emission; HSR, high spontaneous rate; LSR, low spontaneous rate; SSN, speech-shaped noise.
(EPS)

**S5 Fig. Percentage change in TFS encoding for masked speech conditions (BN5/SSN3) after introduction of efferent feedback (15-dB attenuation) to LSR AN fibres (in >1.5kHz frequency band).** Changes to TFS encoding were calculated for masked speech (not for noise-vocoded stimuli given their scrambled fine structure [50,73,76,178]) using Natural conditions with MOC reflex as control templates to calculate both $\rho$TFS$_{AN}$ and $\rho$TFS$_{MOC}$. Adding the MOC reflex produced a mean improvement in TFS encoding for both BN5 and SSN3 (mean $\Delta\rho$TFS for BN5 for freqs >1.5 kHz = 0.31 ± 0.16%, [Z(99) = 2.61, $p = 0.009$, r = 0.26]); mean $\Delta\rho$ENV for SSN3 >1.5 kHz = 1.06 ± 0.15, [Z (99) = 5.95, $p < 0.001$, r = 0.59]). The underlying data can be found in https://doi.org/10.5061/dryad.3ffbg79fw. AN, auditory nerve; BN, babble noise; LSR, low spontaneous rate; MOC, medial olivocochlear; SSN, speech-shaped noise; TFS, temporal fine structure.
(EPS)

**S6 Fig. Cortical evoked potentials during active and passive speech perception.** ERP components during the active and passive listening from electrodes: FZ, F3, F4, CZ, C3, C4, TP7, TP8, T7, T8, PZ, P3, and P4 are shown in panels A, B, and C. Electrode's selection was based on their relevance in attentional and language brain activity related networks [93–97]. Thick lines and shaded areas represent means and SEM, respectively. Within conditions analysis showed that, for all speech manipulations, the magnitude of P1, P2, and N400 potentials were enhanced during active (colour lines) when compared to the passive (grey lines) listening conditions, while N1 tended to be less negative in the active task. LPC magnitude was only significantly enhanced during the active listening of speech in noise. **(A)** ERP components in natural

and all noise-vocoded manipulations: P1: [F (1,24) = 6.36, $p$ = 0.02, $\eta^2$ = 0.21], N1: [F (1, 24) = 16.03, $p$ = 0.001, $\eta^2$ = 0.40], P2: [F (1, 24) = 12.30, $p$ = 0.002, $\eta^2$ = 0.34], N400: [F (1, 24) = 31.82, $p$ = 0.0001, $\eta^2$ = 0.57], LPC: [F(1,24) = 5.29, $p$ = 0.03, $\eta^2$ = 0.18]. **(B)** ERPs during natural (different population than noise-vocoded experiment) and all BN manipulations ($n$ = 29): P1: [F (1, 28) = 24.47, $p$ = 0.0001, $\eta^2$ = 0.47], N1: [F (1, 28) = 10.46, $p$ = 0.003, $\eta^2$ = 0.27], P2: [F (1, 28) = 10.65, $p$ = 0.003, $\eta^2$ = 0.28], N400: [F (1, 28) = 62.16, $p$ = 0.0001, $\eta^2$ = 0.69], LPC: [F(1,28) = 10.55, $p$ = 0.003, $\eta^2$ = 0.27]. **(C)** ERP components during SSN manipulations ($n$ = 29): P1: [F (1, 28) = 22.98, $p$ = 0.0001, $\eta^2$ = 0.45], N1: [F (1, 28) = 6.07, $p$ = 0.02, $\eta^2$ = 0.18], P2: [F (1, 28) = 18.10, $p$ = 0.001, $\eta^2$ = 0.39] and N400: [F (1, 28) = 60.75, $p$ = 0.0001, $\eta^2$ = 0.68], LPC: [F(1,28) = 10.76, $p$ = 0.003, $\eta^2$ = 0.28]. The underlying data can be found in https://doi.org/10.5061/dryad.3ffbg79fw. BN, babble noise; ERP, event-related potential; LPC, late positivity complex; SSN, speech-shaped noise.
(EPS)

**S7 Fig. Comparison of LTAS for natural speech, BN, and SSNs.** Power spectrum density estimates were calculated for 300 concatenated natural speech tokens and 60 seconds of 8-talker BN and SSN; all acoustic stimuli had been normalised to 65 dB for the purpose of this figure. The upper root-mean square envelopes, generated using 300-point sliding windows, are shown for the different conditions. The underlying data can be found in https://doi.org/10.5061/dryad.3ffbg79fw. BN, babble noise; LTAS, long-term average spectra; SSN, speech-shaped noise.
(EPS)

**S8 Fig. Example of subject's CEOAE data management from Fig 1F.** Boxes and whiskers represent the distribution of the data in quartiles. Whiskers indicate the variability outside the upper and lower quartiles. Stars symbols represent outliers, data points labelled **SNR** corresponds to CEOAEs data with snr <6 dB, while data points labelled **ID** corresponds to incomplete data acquisition. These data points were not considered for statistical analysis. The underlying data can be found in https://doi.org/10.5061/dryad.3ffbg79fw. CEOAE, click-evoked otoacoustic emission; SNR, signal-to-noise ratio.
(EPS)

## Acknowledgments

The authors thank Prof. David Poeppel for his contributions during experimental design. We thank Prof. David Ryugo for his comments on the manuscript. In addition, we thank Ronny Ibrahim, Jaime Undurraga, Lindsey Van Yper, and Greg Stewart for their technical support and EEG analysis and Nicholas Clark for his assistance with MAP_BS. The authors thank Ray Meddis for bringing the MAP_BS model to our attention.

## Author Contributions

**Conceptualization:** Heivet Hernández-Pérez, Jason Mikiel-Hunter, David McAlpine, Sumitrajit Dhar, Jessica J. M. Monaghan, Catherine M. McMahon.

**Data curation:** Heivet Hernández-Pérez, Jason Mikiel-Hunter.

**Formal analysis:** Heivet Hernández-Pérez, Jason Mikiel-Hunter, Sriram Boothalingam, Jessica J. M. Monaghan.

**Funding acquisition:** Heivet Hernández-Pérez, David McAlpine, Sumitrajit Dhar, Jessica J. M. Monaghan, Catherine M. McMahon.

**Investigation:** Heivet Hernández-Pérez, Jason Mikiel-Hunter.

**Methodology:** Heivet Hernández-Pérez, Jason Mikiel-Hunter, David McAlpine, Sumitrajit Dhar, Sriram Boothalingam, Jessica J. M. Monaghan, Catherine M. McMahon.

**Project administration:** Heivet Hernández-Pérez.

**Resources:** Heivet Hernández-Pérez, Jason Mikiel-Hunter.

**Software:** Heivet Hernández-Pérez, Jason Mikiel-Hunter, Sumitrajit Dhar, Sriram Boothalingam, Jessica J. M. Monaghan.

**Supervision:** Sumitrajit Dhar, Jessica J. M. Monaghan, Catherine M. McMahon.

**Validation:** Heivet Hernández-Pérez, Jason Mikiel-Hunter, Sriram Boothalingam, Jessica J. M. Monaghan.

**Visualization:** Heivet Hernández-Pérez, Jason Mikiel-Hunter, David McAlpine, Jessica J. M. Monaghan.

**Writing – original draft:** Heivet Hernández-Pérez, Jessica J. M. Monaghan, Catherine M. McMahon.

**Writing – review & editing:** Heivet Hernández-Pérez, Jason Mikiel-Hunter, David McAlpine, Sumitrajit Dhar, Sriram Boothalingam, Jessica J. M. Monaghan, Catherine M. McMahon.

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
