## [Editor Report · Decision Letter 0]

18 Mar 2021

Dear Dr Hernandez Perez, 

Thank you for submitting your manuscript entitled "Perceptual gating of a brainstem reflex facilitates speech understanding in human listeners" for consideration as a Research Article by PLOS Biology.

Your manuscript has now been evaluated by the PLOS Biology editorial staff, as well as by an academic editor with relevant expertise, and I am writing to let you know that we would like to send your submission out for external peer review.

Please re-submit your manuscript within two working days, i.e. by Mar 22 2021 11:59PM.

Kind regards,

Gabriel Gasque, Ph.D.,

Senior Editor

PLOS Biology

---

## [Decision Letter · Decision Letter 1]

27 Apr 2021

Dear Dr Hernandez Perez,

Thank you very much for submitting your manuscript "Perceptual gating of a brainstem reflex facilitates speech understanding in human listeners" for consideration as a Research Article at PLOS Biology. Your manuscript has been evaluated by the PLOS Biology editors, by an Academic Editor with relevant expertise, and by four independent reviewers. Please accept my apologies for the delay in sending the decision below to you.

In light of the reviews (below), we will not be able to accept the current version of the manuscript, but we would welcome re-submission of a much-revised version that takes into account the reviewers' comments. We cannot make any decision about publication until we have seen the revised manuscript and your response to the reviewers' comments. Your revised manuscript is also likely to be sent for further evaluation by the reviewers.

We expect to receive your revised manuscript within 3 months. 

**IMPORTANT - SUBMITTING YOUR REVISION**

Your revisions should address the specific points made by each reviewer. Having discussed these with the Academic Editor, we think reviewer 1 makes a good point about your title. You should be more specific unless you can make a clear demonstration otherwise. You will also note that reviewer 3 suggests you could split this submission into several manuscripts, to decompress the message. We think you should keep all the information within this paper, as its impact relies on the comprehensive collection of data and analyses .

Please submit the following files along with your revised manuscript:

*Re-submission Checklist*

*Published Peer Review*

*PLOS Data Policy*

*Blot and Gel Data Policy*

Sincerely,

Gabriel Gasque, Ph.D.,

Senior Editor,

ggasque@plos.org,

PLOS Biology

REVIEWS:

Reviewer's Responses to Questions

Reviewer #1: This study investigates cochlear, brainstem and cortical responses during passive and active listening monaurally. The active listening task consisted of detecting non-words in a single stream of words and non-words, and the passive task involved presenting similar speech streams while participants were watching and attending to a stop-motion movie. Cochlear and brainstem responses during the two tasks were monitored by recording otoacoustic emissions (OAEs) and auditory brainstem responses (ABRs) to clicks presented to the ear contralateral to speech ear. Cortical responses were obtained by analyzing the P1, N1, N2, N400 and the late positivity component (LPC) after each word. The speech was degraded by noise-vocoding and by adding babble (BN) and speech-shaped noise (SSN).

A main finding is that cochlear responses are inhibited (OAEs suppressed) during active but not passive listening to vocoded speech, while they are inhibited during passive but not active listening to masked speech. This finding is interpreted as indicating that medial olivo-cochlear (MOC) efferents are gated selectively when their activation helps to perform the task, i.e., they are activated when actively listening to vocoded speech because their activation enhances the amplitude fluctuations in vocoded speech, which facilitates understanding vocoded speech; conversely, they are activated in passive listening (during the active visual task) because MOC efferent activation suppresses the speech+noise stimulus that is irrelevant to perform the active visual task. The authors reason that MOC efferents are not active during active listening to masked speech because their activation would not improve the representation of speech in simultaneous noise. To support this interpretation, the authors use a computational model of the auditory brainstem with MOC efferent control. The model predicts better speech-envelope coding with than without MOC efferent activation for vocoded speech but not for masked speech.

Another main finding is that brainstem (wave V) responses are stronger in active than in passive listening to masked speech, while they are not different in the active and passive listening tasks when listening to vocoded speech. Furthermore, in active listening, cortical responses (P1, P2 and LPC) are stronger when listening to masked speech than when listening to vocoded speech. These findings are interpreted as reflecting an engagement of the central auditory system in de-noising masked speech but not when listening to vocoded speech in quiet.

Combined, the findings are interpreted as reflecting that central/cortical auditory areas are involved in de-noising when listening to speech in noise, while MOC efferents are gated during tasks where participants benefit from the effects of MOC efferent activation.

The data are interesting and the interpretations thought-provoking. There are several issues, however, that prevent me from recommending this study for publication in its present form. I would be happy to reconsider my recommendation once the authors ponder (and hopefully address) the following concerns and comments.

Major issues

1.- The authors should be more careful in using the term MOC reflex. The term reflex is conventionally used to refer to an activation of MOC efferents by sound. Therefore, it is confusing to see the same term used to refer to an activation of MOC efferents by active listening (i.e., by attention). Noting the difference in terminology might help reconciling present and past findings. It is conceivable, for example, that MOC-efferent activation by sound (i.e., the MOC reflex) facilitates the neural representation of speech in noise, but attention-gated activation of MOC-efferents facilitates task performance. Therefore, I suggest carefully revising the terminology throughout the manuscript for accuracy and consistency with previous studies.

2.- The manuscript title does not accurately reflect the content and scope of the study and is misleading. As it stands, it implies that the MOC reflex facilitates the understanding of any speech (in noise), when in fact the data do not support this idea. The data support that activation of MOC efferents facilitate understanding noise-vocoded speech in quiet, but not necessarily speech in general, and certainly not speech embedded in noise. I advise revising the title for accuracy.

3.- The description of the stimuli is not clear. It is unclear if words were presented with or without silent gaps between them. Also unclear is whether the BN and SSN were continuous during the 12-minute experimental blocks, including any silent gaps between words, or if they were gated with each word (as implied in L643). Activation of the MOC efferents (and their effects on CEOAEs) would have been very different for continuous and gated noises. Because of the relatively slow time-course of MOC efferent activation (~280-300 ms), if noises were gated with words, and there were silent gaps between words, then MOC efferents would have been active only during the last portion of each word. If my understanding is correct, this was the certainly case for noise-vocoded speech but it is unclear if it was also the case for noisy speech. The temporal course of the various stimuli should be explained more clearly for this reviewer to be able to interpret the results. I suggest including a supplementary figure illustrating more clearly the time course of words, BN or SSN, clicks, and recording windows of OAEs, ABRs, and cortical responses.

4.- The writing needs to be improved. I appreciate the direct and concise writing style, but the manuscript contains a lot of information, some of which is hard to follow without further details. Specific instances where the text can be improved are given below.

5.- The rationale for using the various stimulus configurations is not explained. For example, what was the purpose of (or even better, the hypothesis for) using vocoded speech? Without further details, it seems as though they are building a story backwards from the findings. Please explain upfront the rationale for your stimuli, in particular, the purpose/hypothesis of using vocoded speech in quiet.

6.- The interpretation of the findings is not rounded and not 100% warranted by the data. Why would MOC efferent activation not occur during passive listening to vocoded speech? In this condition, the task was to visually attend to the silent movie and the vocoded speech was irrelevant. Therefore, participants would have also benefited from the MOC system inhibiting the irrelevant auditory stimulus as they benefited (according to your interpretation) from the MOC inhibiting the masked speech in the passive auditory task.

7.- The model support to your findings is not 100% convincing for three reasons. First, it is based on the assumption that the speech understanding is based solely on the neural representation of the speech envelope. This assumption may or may not be true, but people can understand speech reasonably well with only fine structure (Lorenzi et al., 2006, PNAS). Second, if I understood correctly, the correlation between neural and stimulus envelope representations is based on neural responses at one CF only (?), which is unlikely the case. Third, when the same model (or a version of it) has been used in combination with an automatic speech recognizer, the model predicts MOC efferent activation to improve the recognition of speech in noise (Brown et al., 2010, JASA; Yasin et al., 2018, JASA; Yasin et al., 2020, IEEE Access). This latter aspect is discussed in the manuscript but is dismissed by the authors. Altogether, I find that the model and the chosen metric used to predict speech recognition (Delta_rho_ENV) are used in a limited (and somewhat biased) way.

Minor/specific comments

L18. The reflex does not really extract anything, though it might help extracting the salient features in question. Please rephrase for accuracy.

L23. The author(s) of the model reported exactly the opposite, i.e., they reported that MOC efferent activation improved the recognition of speech in noise by an automatic speech recognizer.

L50-51. The end of this sentence appears to be incomplete.

L55. Here and elsewhere, it would be appropriate to cite the review of Lopez-Poveda (2018) Front Neurosci.

L62. Replace the semicolon with a comma.

L63. Remove the space in "OAE s".

L79. I am not sure what mean by "homologous visual and auditory scenes". In what sense were they homologous?

L82. For clarity, I suggest starting this paragraph with "We found that…".

L85. It would be clearer to say "assessed in terms of the suppression of CEOAEs"?

L94. "levels of listening effort". Are you saying (in this paragraph) that listening to vocoded speech involves spending different levels of effort than listening to speech in noise? I would say that it is not only "different levels of" but also "different kinds of" effort?

L101. "to 'natural' speech (i.e., speech-in-quiet)". This is confusing. Consider rephrasing as "to 'natural' speech in quiet".

L105. "Three levels of task difficulty". This description is confusing because, actually, I see more than three levels. Considering rephrasing for clarity/accuracy.

L108. "This was statistically confirmed". What was confirmed?

L113-115. The last statement appears to be incomplete. Please revise.

L120. What test was applied?

L122-123. I am not sure what the statistical test indicated at the end of the sentence refers to.

L132. It would be useful to describe what d' refers to, so that the reader can understand the legend and figure without having to read the Methods. Also, the "white circles" referred to are not visible (or present?) in the figure. Perhaps, this is because of the poor quality (low resolution) of the figures provided to the reviewers.

L160. "auditory efferent activity". Do you mean efferent to the cochlea or more generally?

L162. I am not sure (at this point in the text) how iso-performance was achieved. Please explain.

L168. "(Figure 2A)" I think you probably mean "(Figure 1C)". Please double check and correct it if appropriate.

L193. Delete "and".

L194. "(Figure 1C)" I think you mean "(Figure 1D)". Please double check and correct it if appropriate.

L198. "differing results" to what results are your referring to?

L208. "gain changes are sufficient"; sufficient for what? Please explain.

L215-217. Alright, but other authors have shown this not to be only effect of MOC efferent activation. See my major comments. Why not use, for example, the model in combination with a speech recognizer?

L220-222. Speech recognition is unlikely based on information carried by LSR units? Why not apply the same analyses to all fiber types?

L244. Do you really mean 400 fibers/1ms bin-width or 400 stimulus presentations per fiber? It is not the same thing.

L279-281. How does this compare to the findings of past studies that used the same model? See my general comments.

L282-291 and more generally. Did the masking noise in the model and the experiments start sufficiently well before the words to guarantee full activation of MOC efferents at the time when words started and during the full duration of the words?

L310-316 and more generally. Were the BN and SSN such that the MOC *reflex* (i.e., an activation of MOC efferents by sound) be active during passive listening in noise (Fig. 1C)? 

L320-321. Why are so many electrodes needed to record the responses shown in Fig. 3? Why are responses shown only for the active listening task? What were responses like (i.e., did they show any significant features) in the passive auditory task (active visual task)?

L356. Delete "2016)".

L373. "preserves cochlear gain to prevent deterioration of envelope encoding". Why would activation of MOC efferents deteriorate envelope encoding in quiet or in noise?

L381. Close the brackets after "as speech".

L391. "to maintain iso-performance" � "possibly to maintain iso-performance".

L407. "where participants" � "when participants". (?)

L412-413. "required to attention". Please revise this text.

L417-428. This whole paragraph is hard to follow. Ideas are thrown without indicating which aspects of the data support the ideas in question.

L435. The cited study [62] not only suggests that MOC reflex expands interaural cues associated with sound localization but also; it suggests that the MOC enhances amplitude modulations, and suppresses the noise (and thus enhances the SNR) in the ear with the better acoustic SNR (see for example, Fig. 2 in Lopez-Poveda et al., 2020, Ear Hear).

L469. It would be appropriate to also cite Marrufo-Pérez et al. (2021). Front Neurosci.

L532, "that this analysis" what analysis exactly?

L614. Please define Z.

L643. Was the BN (and SSN) gated with each word? If so, this may have been insufficient for a full activation of the MOC reflex by the noise. Or were the noises continuous? See my general comments.

L682. In what units was CEOAE magnitude expressed? In dB SPL or in Pascal?

L682. CEOAE_baseline (first 60 sec). I am confused. You just said that baseline CEAOE magnitude was calculated over the first and last 60 sec.

L689-694. This (long) sentence appears to be incomplete (missing a verb ?). The MEMR could have been activated by the speech for a significant proportion of participants (see Feeney et al., 2017, Ear Hear).

L696. I am not sure what the 10-20 system is. Please explain.

L698. I am not sure what LSB in nV/LSB refers to. Please define LSB. 

Reviewer #2: The present manuscript presents a large data set to suggest that top-down modulation of the cochlea efferent system depends on the stimulus material, and in particular on how the envelope of speech stimuli is modulated by cochlear gain. This is interesting, as a number of previous studies have suggested top-down control of the efferent system, but the direction of the modulation has been variable, such that the role of this mechanism is not yet well understood. The present manuscript might therefore help to resolve some of the variance between previous studies. The experiments are overall well done and the manuscript is well written. I have only a few questions, which, however, I think are important to interpret the data. As I could not find a certain answer to these in the manuscript, I will discuss them below:

(1) Based on Figure suppl. 1B, there is a 3-s interstimulus interval (ISI) between speech tokens (elsewhere I read that the speech tokens were concatenated?). In the natural speech and NV conditions, I assume the ISI is silent. However, how is the temporal shape of the masker in the other conditions? In the methods section on page 26, it is mentioned that maskers were 60-s long (l. 642), but then also that the speech token and masker segments were matched in duration (l. 643), but no details about onset/offset ramps are provided. Thus, I wasn't sure if the duration of the masker is only of the duration of each speech token, or if it continuous through a 3-s ISI. In the latter case, this would have important influence on the CEOAE, because the ongoing masker would lead to stronger suppression than the silence in the other condition, which would in fact match the data. First, the stimulus setup needs to be explained in more detail (methods refer to Fig Supp 6, but this only includes the spectra). If there are really ISIs that are silent versus filled with a masker, this would need to be considered in the explanation of the data and in the model calculations.

(2) The model suggests that the presence of the MOC reflex enhances envelope coding measured by pENV in the range of 3.5%. How can this effect be compared with CEOAE suppression and modulation of wave V? On first glance the effect size appears relatively small. Could the authors provide some arguments how the effect size could be compared between these meassures?

(3) For the early part of the cortical response, the waveform appears to start right at 0 ms latency, in particular for the conditions where a masker was used. Again, it would be helpful to know more details about the timing of the masker to interpret these waveforms. Is there a common onset of masker and speech token, or is the masker ongoing? (The authors write that "the speech and noise tokens overlap", but this would be the case both with continuous as well as gated maskers). In the case of gated maskers, the onset would represent a mixture of masker and speech token.

Reviewer #3: Review of "Perceptual gating of a brainstem reflex facilitates speech understanding in human listeners" by Hernandez-Perez et al. 

This manuscript describes a study aimed at determining the extent to which Medial Olivocochlear (MOC) reflex activity depends on active/passive listening during the presentation of speech that was intrinsically degraded (i.e., vocoded speech) or degraded by the addition of background noise. The effect of presenting vocoded or noise-degraded speech on the amplitudes of click-evoked otoacoustic emissions (OAEs) and the auditory brainstem response (ABR) was measured by presenting a continuous click train to the non-test ear. Late-latency evoked potentials were also measured in response to the vocoded or noise-degraded speech presented in the test ear. All measurements (OAEs, ABR, late-latency potentials) were obtained simultaneously while the subject actively detected the presence of non-sense words or passively watched a stop-motion movie. The primary findings are 1) suppression of OAEs in response to vocoded speech was significantly larger during active versus passive listening, 2) suppression of OAEs in response to speech degraded by steady-state noise was significantly larger during passive versus active listening, 3) ABR-wave V amplitudes obtained during the presentation of speech-shaped noise were significantly larger during active versus passive listening, and 4) cortical potential amplitudes measured during active listening were larger for noise-degraded speech compared to vocoded speech. The authors interpret a subset of these findings in the context of a computational auditory model where emphasis was placed on analyzing changes in envelope coding for simulations that did versus did not simulate the MOC reflex. 

This study is well designed and appears to produce results that are consistent and are interpretable in the context of previous literature. I appreciate the approach to include simultaneous measurement of perception, OAEs, and event-related potentials and I feel this approach is a great strength to the design of the study. Although I am unable to identify any fatal flaws with the study, I have some general comments that I wish to express to the authors - some of which I give as a matter of opinion and not as changes I require before recommending the manuscript for publication. 

GENERAL COMMENTS

The rationale for the methods surrounding the model simulations is underdeveloped. The manuscript refers to the "Matlab Auditory Periphery and Brainstem" (MAP-BS) computational model; however, a citation is not provided for this model. I was unable to determine 1) whether the authors were proposing a new (unpublished) model, 2) why currently published computational models with efferent feedback were not considered (e.g., Messing et al., 2009; Brown, Ferry, and Meddis, 2010; Smalt et al. 2014), 3) why the MAP-BS model is an improvement over previous models, 4) how the predictions from the MAP-BS model may be similar to or different from previously published models. 

The level of the vocoded and noise-degraded speech was likely intense enough to elicit the middle ear muscle (MEM) reflex. Although the authors measured clinical thresholds for the MEM reflex, there is evidence that clinically-based thresholds are, on average, 18.5 dB higher than more sensitive measures of the MEM reflex based on power reflectance (Feeney and Keefe, 2001). The discussion does not address the likelihood that the stimuli used in the experiment stimulated the MEM reflex; nor does the discussion address how this stimulation modifies the interpretation of the results. 

The manuscript introduces an appreciable amount of new information and findings in a relatively restricted length. I feel this may be overwhelming for the reader. I consider myself an expert in psychophysics, auditory modeling, and MOC efferents, and despite this expertise I found the manuscript overwhelming to read and digest. I feel there are two and maybe three papers stuffed into this one manuscript. For example, if the model simulations are the result of a new (unpublished) model, I think it would be appropriate to dedicate a separate paper to more fully describing the model and the interpretation of the results in the context of the model. Similarly, I am unaware of previous studies that evaluated contralateral suppression of OAEs using vocoded stimuli in passive versus active listening tasks. The discussion of such findings in the context of temporal envelope coding is very dense and may be more digestible if the experimental results were presented in a separate manuscript. The discussion of such a manuscript could more carefully relate these results to suppression of OAEs measured with other stimuli with modulated envelopes and in tasks with/without active listening. I give this final comment as an option for the authors to consider rather than a requirement before I recommend the manuscript for publication. 

Messing, D. P., Delhorne, L., Bruckert, E., Braida, L. D., & Ghitza, O. (2009). A non-linear efferent-inspired model of the auditory system; matching human confusions in stationary noise. Speech communication, 51(8), 668-683.

Brown, G. J., Ferry, R. T., & Meddis, R. (2010). A computer model of auditory efferent suppression: implications for the recognition of speech in noise. The Journal of the Acoustical Society of America, 127(2), 943-954.

Smalt, C. J., Heinz, M. G., & Strickland, E. A. (2014). Modeling the time-varying and level-dependent effects of the medial olivocochlear reflex in auditory nerve responses. Journal of the Association for Research in Otolaryngology, 15(2), 159-173.

Feeney, M. P., & Keefe, D. H. (2001). Estimating the acoustic reflex threshold from wideband measures of reflectance, admittance, and power. Ear and hearing, 22(4), 316-332.

OTHER COMMENTS

Line 28 (abstract): it would help to quickly state/summarize these strategies.

Line 31: I think the goal is not "cocktail-party" listening per se, but effective/robust "cocktail-party" listening.

Lines 50-51: something is wrong with this sentence

Line 57: consider softening the statement by using "unclear" instead of "controversial" 

Line 62: Be more explicit. OAE amplitudes are not always reduced by contralateral acoustic stimulation. In some cases, as with DPOAEs, the amplitude may increase. 

Line 63: There is an extra space between the "E" and "s" of OAEs. 

Line 63-64: This sentence should state that the magnitude of OAE suppression may relate to speech perception performance. (not just OAE magnitude).

Line 64-65: Efferent modulation of cochlear gain could depend…? (I'm not sure what is meant by "confounding effects.")

Line 108: State what "this" is for clarity. (e.g., This "modulation of task difficulty" was statistically…)

Lines 155-156: Why did the direction of the t-statistic change? Shouldn't all t-statistics be in the same direction if CEOAE suppression is present?

Lines 179-180: "Effects" is vague. Do you mean intermediate magnitude of suppression of OAEs?

Lines 207-210: Or alternatively, the changes in ABR are inherited from the cochlear reduction in gain in the noise-degraded speech conditions (brainstem passively transmits signal from lower auditory centers), and for vocoded stimuli there is a compensatory central amplification to offset the reduction in cochlear gain.

Lines 485-488: This conclusion may depend on SNR. For favorable SNRs the expansion would enhance the effective speech envelope while releasing AN fibers from adaptation/saturation produced by the noise.

Lines 496-501: This sentence is long and hard to read. Consider breaking it apart. 

Line 597: State that all subjects did not complete all experimental measurements.

Line 599: Close parentheses after "tympanometry." 

Line 600: Close parentheses that started before "assessed" on line 599. 

Line 600-601: Something is wrong with this sentence.

Line 620: Why was the speech material only from one talker? Please address whether the results of the experiment would be different if a different talker were used. Can we be confident that the results would be similar with a different talker? 

Line 729: Why were the model simulations limited to these frequencies? I expected the simulations would include a broad range of CFs given that the stimulus of interest (speech) is broadband. 

Line 750: The model is not fully described in the text. For example, does the model simulate basilar membrane nonlinearity (i.e., suppression, compression, level-dependent tuning)? I assume it does, but I could not find the description in the text. 

Reviewer #4: Hernandez-Perez et al review

This study uses a clever experimental design with simultaneous OAE, ABR and speech EEG recordings to investigate how cochlear gain changes when (modified) speech is presented in attended vs unattended listening. The topic is very relevant and timely, because little is known about how bottom-up and top-down mechanisms alter the MOC reflex when listening to (degraded) speech. This study is important in that it evaluates cochlear, brainstem and late potential changes simultaneously, which is a rare but necessary approach to understand the cascade of peripheral and neural adjustments the MOC reflex can inflict. In the study, CEOAE gain changes are interpreted as MOC reflex changes that appear to be modulated by the speech modifications and listening state. Model simulations with MOC reflexes on or off can capture the effect of how the stimulus modifications affect the envelope similarity between the natural-MOC condition. The paper ends with a theoretical interpretation of how the trends in the data can be interpreted in a model of (attention-driven) top-down and bottom-up MOC modifications that depend on the acoustics. 

Major comments:

The analysis of the results is of very high quality. However, I think that the connection with the ABR results can be made stronger and that the model/interpretation proposed is not necessarily the "sole" or "conclusive" answer as to how these results could have arisen. At the same time, there is a methodological reference issue (baseline for CEOAE was different from that in modeling) that may have had an impact on the results interpretation. This would have to be sorted out before we can fully grasp the meaning of the results and evaluate whether the proposed models are appropriate. It is rather a collection of aspects that make me doubt whether the data observations warrant the final conclusions, but hopefully these can be addressed/mitigated/clarified in a revision. 

The below comments are in random order of importance:

Minor spelling:

L33: its => their

L46: check em-dash and bracket positions

L51: check bracket text and references

L58: "the/a" listening task

L63: OAEs

L120: "three-way" ANOVA?

L168: should be figure 1, no?

L180: CEOAEs

L193: "and was", change into "- was"

L194: Figure 1D?

L199: Vs. => vs.

L202-205: Question related to testing intrinsic differences in the two populations, and inference of conclusion: "the differences observed in auditory brainstem/midbrain activity and cochlear gain can be attributed to the presented speech degradation when there is no relationship between CEOAE suppression and ABR amplitudes in the natural speech condition" L202-205 show results of a t-test where wave-III/V amplitudes [in microV] are compared to CEOAE suppression [in dB] to support this statement statistically. Does it make sense to compare dB values to microV amplitudes, they have different units and why not characterize ABR reduction rather than amplitude to have a meaningful comparison? At the moment, the 0-hypothesis of your t-test is: "there is no difference in the distributions: dB suppression vs ABR amplitude in microV". To me, such a comparison makes no sense at all, there might never be a theoretical relationship between these two units. Have I misunderstood something about the purpose of the test you conducted, or can you use the same units? 

Figure 1C-D, It is confusing to see CEOAE suppression in negative dBs, are these just small values or are you observing a "release from suppression", i.e. more gain in the system when more background noise is presented? 

L206-210: It is hard to follow your conclusion here, you would need to motivate this better. When I look at the passive conditions only for CEOAE suppression (Figure 1C), and compare the two natural conditions, there is quite some variability between the degree of CEOAE suppression, what is the reason for this, and is this meaningful? My worry is that the BN10/BN5 conditions show more suppression than the corresponding natural condition, but not to the natural condition that came with the vocoded manipulations. Hence it may be difficult to conclude the "reduction of the cochlear gain for masked stimuli" part of your sentence. You then go on to write that the "ABR magnitudes appear only inherited from [this cochlear gain reduction]". Again, only looking at the passive conditions (Figure 1D), I can see that wave-V is reduced for the BN & SSN compared to the vocoded (significance not sure) conditions, but you do not compare against the "Natural" condition. This collapsing in Figure 1D is confusing to me, at least you'd have to isolate the "Natural" conditions. L208-210 are even more difficult to follow, it is not clear how exactly the results lead you to conclude this.

L228: Here, logically you use the "natural speech" as the template for comparison with the iso-response conditions, why not do the same for the CEOAE and ABR conditions? There you are referring to the control as the condition without speech. It would be more consistent to use the same reference throughout the paper. In consequence, CEOAE suppression might become positive in some cases, but it would make it easier to judge the effect of speech manipulations, attention and masking if you used the natural speech as reference in all cases.

L236: It is unclear to me what you mean by "to their natural speech controls" in this section, because earlier on you write "Natural speech control simulations were always performed with the MOC reflex". From a visual inspection of Fig.2A, where this distinction does not matter, VOC8 resembles well the corresponding Nat (compare MOC with MOC and AN only with AN only). The BN/SNN matches less well the corresponding natural conditions (compare MOC with MOC, and AN with AN). The MOC on or off does not change much to these trends as long as you use the corresponding Nat condition. So yes, VOC shows increased similarity to Nat than BM/SNN, but this appears MOC independent. Adding MOC does have an effect on all conditions, but from a visual inspection, but the trends are similarity in comparison to the Nat condition appear the same. Perhaps you did refer to corresponding controls for these comparisons, but it is not clear to me from the description, and the used reference may have a big influence on the results interpretations.

Figure 1 D and text: Also here I am confused as to whether �ENVAN on the x-axis uses (a) the corresponding natural or (b) always MOC natural as reference. In case of (a) you can safely conclude that adding MOC has an effect on envelope similarity. In case of (b) you cannot because clearly the MOC simulations will be more similar to the MOC natural than the no-MOC simulations, especially for the VOC conditions. So in case you use (b), the sentence "with largest enhancements observed for words with the lowest �ENV values in the absence of MOC reflect" appears trivial and a consequence of always using MOC natural as the reference for both MOC and no-MOC manipulated speech conditions. 

Model Simulations: To better integrate the ABR results into the resulting final proposed mechanistic MOC reflex model, it would be worthwhile to also simulate a proxi for ABRs with the Meddis model. You can sum up the simulated AN responses across fibers and CFs in the to evaluate how the click ABR would be affected with and without MOC, and see how these alterations correspond to the amplitude changes you observed experimentally. Of course you'd only have a reliable wave-I simulation from that model, but you could add a basic-non gain CN/IC model to simulate the ABR wave-III and V as well (e.g. Nelson and Carney, 2005, Verhulst et al., 2018). It is not problematic to me if you decide not to pursue ABR simulations all the way to the wave-V, but even simulating wave-I would already be informative to better understand the bottom-up MOC strength changes and their effects on expected click ABR apmplitude changes (that you also measured). 

Model Simulations/Data interpretations: On the one hand you state in L671 that you analyzed the rms suppression of CEOAE only in the 1-2 kHz region because that is where the MOC reflexes are strongest. On the other hand, you consider envelope tracking as a measure of speech coding and study effects of on-CF MOC simulations. Does this not bias your interpretation, i.e. you simulate MOC effects for speech/noise frequencies outside the 1-2 kHz region as well, where no MOC innervation is expected based on CEOAEs. I am assuming that speech envelope tracking parameters mostly gets their contributions from high-frequency coding (i.e. from ANFs that do envelope tracking with CFs above the phase-locking limit), how do you reconcile these two aspects, and minimize bias on your results interpretation? 

Figure 2E "CEOAE inhibition" => use a similar nomenclature in Fig 1 and 2 when referring to the CEOAEs.

Methods: 

When recording simultaneous EEG and OAEs, it is possible to generate stimulus artifacts in the EEG traces caused by the ER-10B OAE mic. Did you observe such artifacts, and how did you ensure they had no confounding effect on your ABR amplitudes?

Did you only use only one observer to peak-pick your ABR traces (unusual in audiological studies), and did these amplitudes correspond to wave-III and V peaks to noise-floor or positive peak to negative peak amplitudes?

How did you minimize the effects of individual CEOAE/MOC strength on your recordings? In our own recordings, we tend to observe that people with strong CEOAEs shows the most suppression. This is not unusual because they have the most gain in their CEOAEs to begin with. Could individual differences in CEOAE amplitude and MOC strength have had an impact on your results?

---

## [Decision Letter · Decision Letter 2]

28 Sep 2021

Dear Dr Hernandez Perez,

Thank you for submitting your revised Research Article entitled "Understanding degraded speech leads to perceptual gating of a brainstem reflex in human listeners" for publication in PLOS Biology. I have now obtained advice from original reviewers 1 and 4 and have discussed their comments with the Academic Editor. 

Based on the reviews, we will probably accept this manuscript for publication, provided you satisfactorily address the remaining points raised by the reviewers. Please also make sure to address the following data and other policy-related requests.

(1) Please provide a blurb, which will be included in our weekly and monthly Electronic Table of Contents, sent out to readers of PLOS Biology, and may be used to promote your article in social media. The blurb should be about 30-40 words long and is subject to editorial changes. It should, without exaggeration, entice people to read your manuscript. It should not be redundant with the title and should not contain acronyms or abbreviations. For examples, view our author guidelines: https://journals.plos.org/plosbiology/s/revising-your-manuscript

(2) Please indicate within your manuscript if your experiments were conducted according to the principles expressed in the Declaration of Helsinki or any other (specific) national or international ethical guidelines.

(3.a) We thank you for providing a Reviewer Dryad URL with your data. If you are going to use Dryad as your data repository, please make sure the link goes live before acceptance.

(3.b) Please include in your data folder/Dryad URL a README file that explains how the source data were analyzed to generate the graphs displayed in the figures.

(3.c) Please double check your Excel file for S2 Fig and S5 Fig because the “tabs” seem to be mislabeled.

(3.d) Please include in each figure legend a sentence indicating where the underlying data can be found. For example, you can write, "The underlying data can be found in [Dryad URL]".

We expect to receive your revised manuscript within two weeks. 

*Published Peer Review History*

*Early Version*

Sincerely,

Gabriel Gasque, Ph.D.,

Senior Editor,

ggasque@plos.org,

PLOS Biology

Reviewer remarks:

Reviewer #1: The authors have considered each and every one of my comments and have addressed all of them adequately. The revised manuscript is much clearer and stronger. The study is very interesting and timely. It will be of interest to hearing scientists and audiologists. It should be published pending the following comments.

General comment

One general comment is that the notion that the MOC reflex can help understanding vocoded speech because it enhances the amplitude modulations in speech was suggested by Lopez-Poveda et al. (2016). Indeed, it is explicitly shown in the insets of Fig. 2 of a more recent study by the same group (Lopez-Poveda et al., 2020, Ear and Hearing). Of course, their conclusions are based on the behavior of a MOC-inspired sound-coding strategy for cochlear implants. It is nice that the present study provides experimental support to that notion also for normal-hearing listeners. What I find puzzling is that Lopez-Poveda et al. (2020) found that the effect and benefit apply to (vocoded) speech in quiet as well as in noise, while the present study find the effect and benefits to apply only in quiet. I would ask the authors to consider highlighting and discussing the issue briefly in their manuscript.

Minor comments

L107. Remove the comma after "different kinds,".

L152-153. Do whiskers really represent interquartile range (IQR=q3-q1)? Please double check because this does not seem to be the case for some boxplots, e.g., Natural and Voc16 in the left panel.

L751. Delete the blank space in "to increase ."

L775. Delete the comma in "each time, they heard".

L793. Delete the blank space in "stop/ non-stop".

L812-813. Replace the full stop with a comma in "applied to stimuli. Figure 1C".

Figure 2D. Please double check the titles for the panels; they seem incorrect. Specifically it makes no sense to write "Nat_MOC vs Nat_MOC" or "VOC_AN vs VOC_AN".

Figure 3A. Correct the ordinate label in the right-top panel; it reads "nplitude" where it should read "Amplitude".

Reviewer #4: Revision: 

Understanding degraded speech leads to perceptual gating of a brainstem reflex in human listeners.

Hernandez-Perez, Mikiel-Hunter et al.

I would like to congratulate the authors on their revised manuscript. It follows a unique whole-systems approach to unravel the functional role of the MOC reflex in (degraded) speech perception, a topic that is both timely and important. I appreciate the careful consideration of the reviewer comments/suggestions and subsequent additional analyses, which substantially improved the clarity of the manuscript and led to a stronger scientific argumentation. The editor asked me to also evaluate rebuttal comments to R2, and I can confirm that you addressed both our raised points satisfactory. I have one comment left that you may wish to consider in your final submitted version (no need for another rebuttal on my behalf).

L233: "We conclude from this that the differences observed in cochlear gain and auditory brainstem/midbrain activity can be attributed to the specific form of speech degradation. Together with the effect on CEOAEs, our data suggest that the magnitude of auditory midbrain activity for the different speech manipulations reflects cochlear output. While this is evident for both listening conditions in masked speech, the similarity of ABR amplitudes in the midbrain for active and passive listening of noise-vocoded stimuli is indicative of feed-forward amplification that compensates for reduced cochlear gain during active listening"

I understand how you get to your conclusion, but I find the direct comparison between CEOAE and ABR suppression as motivator for the "feed-forward amplification" explanation a bit tricky. It is a possibility, but you need to also consider that ABRs and CEOAEs reflect activity from somewhat different cochlear regions. Dominant CEOAE energy in 1-2 kHz, whereas regular click ABRs mostly reflect summed activity of the higher frequency regions (3-8 kHz, Abdala & Folsom 1995, Don & Eggermont 1978). Can you be sure that this latter aspect had no influence on your interpretation of results?

L372-374: consider rephrasing for clarity

References

Analysis of the click-evoked brainstem potentials in man using high-pass noise masking

M. Don, and J. J. Eggermont The Journal of the Acoustical Society of America 63, 1084 (1978); doi: 10.1121/1.381816

Frequency contribution to the click-evoked auditory brain-stem response in human adults and infants. Carolina Abdala, and Richard C. Folsom. The Journal of the Acoustical Society of America 97, 2394 (1995); doi: 10.1121/1.411961 View online: https://doi.org/10.1121/1.411961

---

## [Editor Report · Decision Letter 3]

7 Oct 2021

Dear Heivet,

On behalf of my colleagues and the Academic Editor, Manuel S. Malmierca, I am pleased to say that we can in principle offer to publish your Research Article "Understanding degraded speech leads to perceptual gating of a brainstem reflex in human listeners" in PLOS Biology, provided you address any remaining formatting and reporting issues. These will be detailed in an email that will follow this letter and that you will usually receive within 2-3 business days, during which time no action is required from you. Please note that we will not be able to formally accept your manuscript and schedule it for publication until you have made the required changes.

PRESS

Sincerely, 

Gabriel Gasque, Ph.D. 

Senior Editor 

PLOS Biology

ggasque@plos.org